

# Ice nucleating particles over the Eastern Mediterranean measured by unmanned aircraft systems

Jann Schrod[1], Daniel Weber[1], Jaqueline Drücke[1], Christos Keleshis[2], Michael Pikridas[2], Martin Ebert[3], Bojan Cvetković[4], Slobodan Nickovic[4], Eleni Marinou[5,6], Holger Baars[7], Mihalis Vrekoussis[2,8,9], Nikos Mihalopoulos[2,10], Jean Sciare[2], Joachim Curtius[1], and Heinz G. Bingemer[1]

[1]Institute for Atmospheric and Environmental Sciences, Goethe University Frankfurt, 60438 Frankfurt am Main, Germany
[2]Energy, Environment and Water Research Center, The Cyprus Institute, Nicosia, 2121 Aglantzia, Cyprus
[3]Institute for Applied Geosciences, Technical University of Darmstadt, 64287 Darmstadt, Germany
[4]Republic Hydrometeorological Service of Serbia, 11000 Belgrade, Serbia
[5]Institute for Astronomy, Astrophysics, Space Applications and Remote Sensing, National Observatory of Athens, 15236 Athens, Greece
[6]Department of Physics, Aristotle University of Thessaloniki, 54124 Thessaloniki, Greece
[7]Leibniz Institute for Tropospheric Research, 04318 Leipzig, Germany
[8]Institute of Environmental Physics and Remote Sensing - IUP, University of Bremen, 28359 Bremen, Germany
[9]Center of Marine Environmental Sciences - MARUM, University of Bremen, 28359 Bremen, Germany
[10]Institute for Environmental Research and Sustainable Development, National Observatory of Athens, 15236 Athens, Greece

*Correspondence to:* Jann Schrod (schrod@iau.uni-frankfurt.de)

**Abstract.**

During an intensive field campaign on aerosol, clouds and ice nucleation in the Eastern Mediterranean in April 2016, we have measured the abundance of ice nucleating particles (INP) in the lower troposphere from unmanned aircraft systems (UAS). Aerosol samples were collected by miniaturized electrostatic precipitators onboard the UAS at altitudes up to $2.5\,\mathrm{km}$.

5 The number of INP in these samples, which are active in the deposition and condensation modes at temperatures from $-20$ to $-30\,^{\circ}\mathrm{C}$, were analyzed immediately after collection on site using the ice nucleus counter FRIDGE. During the one month campaign we encountered a series of Saharan dust plumes that traveled at several kilometers altitude. Here we present INP data from 42 individual flights, together with aerosol number concentrations, observations of lidar backscattering, dust concentrations derived by the dust transport model DREAM (Dust Regional Atmospheric Model), and results from scanning electron

10 microscopy. The effect of the dust plumes is reflected by the coincidence of INP with the particulate mass (PM), the lidar signal and with the predicted dust mass of the model. This suggests that mineral dust or a constituent related to dust was a major contributor to the ice nucleating properties of the aerosol. Peak concentrations of above $100\,\mathrm{INP\,std.l^{-1}}$ were measured at $-30\,^{\circ}\mathrm{C}$. The INP concentration in elevated plumes was on average a factor of 10 higher than at ground level. Since desert dust is transported for long distances over wide areas of the globe predominantly at several $\mathrm{km}$ altitude we conclude that INP

15 measurements at ground level may be of limited significance for the situation at the level of cloud formation.



# 1 Introduction

Ice nucleating particles (INP) act as a seed-surface for water vapor and liquid water to enable the emergence and growth of ice crystals in the atmosphere. The process of ice nucleation can occur by immersion, condensation, deposition or contact freezing (for a detailed explanation see: Vali et al., 2015). Despite their low abundance in the atmosphere, INP are crucial for the

evolution of ice in clouds. Without their existence the ice phase in clouds would solely arise from homogeneous ice nucleation at temperatures below about $-36\,°C$. In the presence of INP water freezes at much higher temperatures by heterogeneous ice nucleation, thus affecting the formation of clouds, precipitation and climate. It is a well-known fact that the ice phase plays a key role in the development of precipitation via the Wegener-Bergeron-Findeisen process. Mülmenstädt et al. (2015) found that precipitation especially over continental regions and the mid-latitude oceans is mainly produced via the ice phase. However, the

abundance and distribution of INP are largely unknown (DeMott et al., 2010). Partly due to this fact, the estimates of radiative climate forcing presented in the IPCC AR5 show the largest error bars and the lowest levels of confidence in the category of cloud adjustments due to aerosol (IPCC, 2013).

 Many topics in ice nucleation are not yet investigated to a satisfying degree: e.g. the abundance and vertical profile of INP as a function of cloud nucleation conditions (temperature and supersaturation), the aerosol particle spectrum, as well as the nature,

source and properties (such as morphology, chemical composition and degree of coating) of individual INP. Which physical and chemical properties favor the formation of ice on the surface of an aerosol particle still remains a field of intense research. Furthermore, field data of INP are very limited in a number of ways. The number of observations is still relatively small. These observations cover only a few selected geographical locations and the bulk is coming from the United States of America or Europe. Very few to no measurements are available for the vast majority of other regions of the world (or even whole continents

and oceans). Although from laboratory experiments a number of aerosol species are identified to nucleate ice, still rather little is known on their importance in the real atmosphere. Primary biological particles such as certain bacteria like *pseudomonas syringae* show a high potential to nucleate at high temperatures (Schnell & Vali., 1973; Maki et al., 1974; Vali et al., 1976; Wex et al., 2015). However, little is known on their atmospheric abundance, which appears to be low. Whether or not they have an actual global scale impact on ice-formation in mixed-phase clouds is still unclear (Hoose et al., 2010). Recently, O'Sullivan

et al. (2015) proposed that nanometer-scale fragments of biogenic particles, which appear to be far more numerous than the supermicron parent species, may indeed play an important role for cloud glaciation processes, especially at temperatures above $-20\,°C$. While these biological nanoparticles are unlikely to be directly aerosolized in great numbers from the surface, they may be mobilized mixed together with soil dust particles. O'Sullivan et al. (2016) demonstrated that nanoscale particles from the common fungus *Fusarium avenaceum* adsorb easily to kaolinite, and transfer their high ice-nucleating activity to the clay

in the process. They found that even after multiple water washings the mixture preserved a high ice-activity. Yet, the absolute source strengths of such nanometer-scale biological particles, released into the atmosphere, still needs to be determined.

 On the other hand, mineral dust particles originating from the deserts have been postulated since long ago (Wegener, 1911) to be major contributors to atmospheric INP. Mineral dust particles have been consistently identified worldwide as INP in many studies (e.g. Kumai, 1951; Isono, 1955; DeMott et al., 2003b; Pratt et al., 2009; Prenni et al., 2009b), usually at temperatures



below $-20\,°C$. Many of the earlier field studies conclude the ice-nucleating properties of dust by circumstantial reasoning, i.e. from correlation of dust parameters with INP abundance. The advanced coupling of INP measurements to electron microscopy or to mass spectrometry, which allows identification of single nucleating particles, again demonstrated that mineral dust is a major constituent of INP, but that a significant biological INP-compound can be present too (Pratt et al., 2009; Prenni et al., 2009b). INP concentrations measured within a dust layer can reach up to $1000\,l^{-1}$ (DeMott et al., 2003a). Once mobilized by high surface winds in arid and semi-arid regions of the world, mineral dust particles can travel up to several thousands of kilometers (Prospero, 1999). Hence, regions that are far away from the desert can still be influenced by mineral dust due to efficient long distance transport. Liu et al. (2008) have shown the first height-resolved global distribution of dust aerosol based on lidar measurements of CALIPSO. They found northern hemispheric spring to be the most active dust season, with 12 % of the areas between 0° and 60° N to be influenced at least half the time. In general the vertical extent of the mineral dust was found to be strongly dependent on season and source region. Peak dust layers were found to be between 2–3 km in summer and 1–2 km in the other seasons. The regions of North Africa and the Arabian Peninsula were found to be the most persistent sources of mineral dust. A transatlantic transport of African dust was seen all year, with a significant amount of dust transported in the free troposphere in layers above 2 km in summer. In winter most dust was transported below 2 km. In summary, existing climatologies demonstrate that mineral dust is mainly transported in the lowest few kilometers of the atmosphere. Nonetheless, to this date most INP measurements are made at ground level and not at elevations where clouds actually are formed.

These findings strongly emphasize the need for more measurements of INP above ground level. Yet, achieving INP measurements in free tropospheric air masses is challenging and usually requires substantial effort. For over a decade large research aircraft equipped with continuous-flow diffusion chambers (CFDC) have been used to measure INP concentration and composition in the free troposphere (e.g. Rogers et al., 2001a, b; DeMott et al., 2003a; Cziczo et al., 2004; Prenni et al., 2009a). So far, a light-weight and easy-to-use alternative solution has been lacking. The unmanned aircraft systems (UAS) based offline INP measurement technique presented in this study may be able to fill this gap. However, admittedly, this technique lacks the capability of high time resolution and large spatial coverage as can be expected with a CFDC/aircraft combination. With their easy commercial availability, their high flexibility and light dimensions, UAS theoretically offer the potential of more frequent or even regular measurements of INP in many different locations all around the world. This could enhance the number of global INP observations drastically, therewith shedding light on many regions of the world where such data is missing. While the UAS used here could not be operated in the altitude regions where it is cold enough for INP activation to take place, our free tropospheric measurements are likely to be much more representative of the atmospheric conditions of ice formation in clouds than ground based measurements. Larger drones (e.g. global hawk) might one day even be offering the space for online measuring CFDCs, although many of the advantages of the light and easy setup of smaller sized UAS would be lost.

The scientific community in various fields has already identified UAS technology as a great platform for new approaches. The diverse areas of investigations with UAS cover remote sensing (Colomina et al., 2014; Watts et al., 2012), meteorological profiling (Holland et al., 1992; Reuder et al., 2009) or greenhouse gases (McGonigle et al., 2008), to name a few. More recently, vertical profiles and distributions of aerosol particles (Corrigan et al., 2008; Bates et al., 2013) and cloud microphysical parameters (Roberts et al., 2007; Jensen et al., 2012) have been measured with UAS, highlighting the versatility and the immense



potential of UAS-operated observations, especially for atmospheric sciences. However, to our knowledge, no measurements of INP based on UAS have been performed or published to this day.

The method we present here to measure INP from the UAS is based on the offline ice nucleus counter FRIDGE (Schrod et al., 2016). FRIDGE couples the sampling of INP on board the UAS by electrostatic precipitation of aerosol particles onto

substrates with the subsequent analysis of the substrates in the isostatic diffusion chamber FRIDGE. The presence of INP is detected by growing ice crystals on the substrate. No information on the nature of the INP is derived. The question whether the ice nucleation happened on a dust particle or a biological particle mixed with the dust remains open. An advanced version of our method that couples INP detection by FRIDGE to subsequent electron microscopy analysis of the individual active INP was not applied here, due to the heavy loading of the samples.

From March 27 to April 28, 2016, a joint field campaign of Ice Nuclei Research Unit (INUIT) and the EU projects BAC-CHUS (Impact of Biogenic versus Anthropogenic emissions on Clouds and Climate: towards a Holistic UnderStanding) and ACTRIS (Aerosols, Clouds, and Trace gases Research InfraStructure) took place at a remote location of Cyprus in the Eastern Mediterranean Sea, an environment that is frequently affected by desert dust. The campaign objectives were centered on studying aerosol and cloud properties, with a special interest in ice nucleation. This study is the first in a series of publications to

come from this experiment. For more information about the campaign itself and a more detailed analysis of the meteorological situation we refer to an upcoming overview paper, which will also cover simultaneous ground-based measurements of INP by other groups and instruments (Kanji et al., in preparation).

## 2 Methods

### 2.1 Site description and campaign setup

The island of Cyprus is located in the Mediterranean Sea approximately 100 km south of the Turkish mainland, 100 km west of the Syrian coast and 300 km north of the Egyptian border. This geographical location with the close proximity of the Sahara Desert in the southwest and the deserts of the Arabian Peninsula in the southeast favors a regular encounter of mineral-dust-rich air masses. Particularly during spring time and early summer, depressions south of the Atlas Mountains mobilize dust and inject it into the westerly flow (Moulin et al., 2008; Querol et al., 2009; Pey et al., 2013). Israelevich et al. (2001) identified

the Chad basin at 16° N, 16° E and the Eljouf basin in Mauritania at 5° W, 19° N as the major source areas of dust for the Mediterranean basin. Furthermore, Cyprus is influenced by marine aerosol and by anthropogenic emissions from South-Eastern Europe and the Middle East. The exposure to these highly variable emission sources favored our choice of this location for an intensive campaign on aerosol properties and ice nucleation. During the campaign three independent measurement sites were operated: Lidar measurements were performed in Nicosia (35°08'26"N, 33°22'52"E, 181 m asl). At the Cyprus Atmo-

spheric Observatory (CAO) on the foothills of Troodos Mountains (Agia Marina Xyliatou, 35°02'19"N, 33°03'28"E, 532 m asl, www.cyi.ac.cy/cao/) aerosol properties and INP were measured. The UAS based observations were carried out 6.5 km north of CAO at the Cyprus Institute UAS airfield (Orounda, 35°05'42"N, 33°04'53"E, 327 m asl, www.cyi.ac.cy/index.php/usrl.html). This paper focuses on the INP measurements conducted on the UAS.



## 2.2 Meteorological conditions

During the campaign the Eastern Mediterranean was mainly under westerly flow, as indicated by the contours of the monthly mean stream function at 500 hPa (Fig. 1a). This flow meandered according to the eastward propagation of troughs and ridges with periods of several days. At sea surface level, pressure gradients over the Eastern Mediterranean were mostly weak (Fig. 1b). The days of March 27 to 30 in Cyprus were characterized by cold, cyclonic conditions, and April 1 to 8 by warm, anticyclonic conditions. A low pressure system that was cut off the major trough over Spain on April 6 traveled slowly eastward along the North African coastline and the Eastern Mediterranean Sea and over Cyprus (April 12 to 13) towards Syria, where it dissipated. On April 14 to 20 anticyclonic conditions prevailed in Cyprus with westerly flow of warm air, on April 21 to 27 again a cyclonic pattern with predominantly warm southwesterly air was present. Supplement S1 gives a more detailed view on atmospheric transport as seen by the DREAM model (cf. sections 2.8 and 3.1), using the dust load as a tracer for atmospheric motion during the campaign. Figure 2 shows the meteorological conditions at CAO during the campaign. During the first two weeks of the campaign the daily maximum temperature was increasing and relative humidity was decreasing. This was followed by a strong increase of humidity on April 9 and a period of high humidity and lower temperatures with a few millimeters of rain on April 12. Days with rising temperatures and decreasing humidity conditions then followed. The local wind direction at CAO was nearly 70 % of the time from the western sector with wind speeds usually below 5 m s$^{-1}$ (Figs. 2c and 3). Figure 4 shows the 10-day backward trajectories ending at 1000 m above ground at Orounda. The model used here is the Hybrid Single Particle Lagrangian Integrated Trajectory Model (HYSPLIT, Stein et al., 2015; Rolph, 2016). Trajectories were calculated at 03:00, 06:00 and 09:00 UTC, in phase with most UAS flights which took place from 04:00 to 12:00 UTC. The vast majority of air masses that reached the UAS site were transported from the west over the Mediterranean Sea. More than 10 % of the trajectories touched Central and Western Europe, northern Africa and the Atlantic. Additionally, more than 5 % of the trajectories showed paths over Northern Europe and the northern Atlantic or the African continent. Since many trajectories originated from the Sahara or adjacent regions, mineral dust particles were episodically transported to Cyprus during the campaign. Fig. 5 displays the aerosol mass concentration (particulate matter, PM) during the campaign. In desert dust that was transported over thousands of kilometers the particles usually still have a diameter of a few microns (e.g. Prospero, 1999) and thus are larger than most other aerosol species. As these large particles make up the bulk of aerosol mass, PM can be considered as a good proxy for mineral dust in the air. By far the highest concentration of PM occured on April 9.

## 2.3 Unmanned Aircraft Systems

Two different types of UAS were used for INP sampling in this campaign. They are described below.

The Cruiser (Fig. 6) is a fixed-wing, medium-size UAS (3.8 m wingspan) with a two-stroke engine and a maximum take-off weight of 40 kg that can carry a payload of up to 10 kg for a maximum flight duration of 3 hours. During the campaign the maximum altitude never exceeded 2.5 km above ground level (ca. 2850 m asl) due to flight plan restrictions. Flight duration was approximately 1.5 hours.





The Skywalker X8 (Fig. 7) has a wingspan of 2.1 m, an electric engine and a maximum take-off weight of 5 kg. It can fly up to 3 km altitude with an endurance of about 1 hour. The maximum altitude reached in this campaign was 2.5 km above ground level with a maximum flight duration of about 1 hour. Compared to the Cruiser it is a much more flexible system as it does not require a runway for take-off and landing. Indeed, it can take off from almost anywhere using a bungee-launching catapult system and it lands on its belly. On the other hand, it is limited to a payload of ca. 2 kg.

The flights were operated from a mobile ground control station, which is equipped with state-of-the-art technology to establish a stable communication link with the UAS for reliable data streaming. The UAS airfield site consists of a private paved runway of $12 \times 200$ m and overhead airspace, which is approved by the Civil Aviation Authority of Cyprus. Both UAS run on the same autopilot system, which enables the UAS to fly automatically in a pre-programmed flight plan.

## 2.4 Measurements of ice nucleating particles: FRIDGE

INP concentrations were measured by electrostatic precipitation of aerosol particles onto silicon wafers onboard the UAS, followed by laboratory analysis of the samples in the ice nucleus counter FRIDGE. The method for INP measurements has been originally introduced in Bundke et al. (2008) and Klein et al. (2010), and was re-evaluated by Schrod et al. (2016). FRIDGE usually addresses the deposition/condensation freezing mode(s). FRIDGE's aerosol sampling unit (programmable electrostatic aerosol collector: PEAC; Schrod et al., 2016) either starts and stops sampling automatically on a prescribed time window or can be remotely controlled, when connected to a local network or the internet. Sampling procedure, transportation and storage of wafers are easy to handle without strict precautions, making it a well-suited instrument for the use in an UAS.

Both UAS were equipped with a customized inlet system connected to an aerosol sampling unit. A custom build, light weight version (600 g) of a single-sampling PEAC was integrated in the Skywalker X8 UAS. The Cruiser UAS had a 2.5 kg multi-sampling PEAC installed, which enabled the sampling of up to seven substrates in one flight. Thereby an altitude profile could be sampled from a single flight. The PEAC uses the principle of electrostatic precipitation to ensure that aerosol particles are homogeneously distributed on the silicon sample substrate. In the sampling process, a pump is used to establish a constant air flow (5 l min$^{-1}$), a high voltage source generates an electric field and the aerosol particles are charged negatively by collisions with electrons from a corona discharge. The charged aerosol particles then precipitate on the grounded silicon wafer. The sampling process was controlled remotely from the ground control station and was started as soon as the UAS reached the desired altitude and flight conditions were stable. Sampling length was selected according to criteria like the presence of dust, aerosol concentration, weather conditions, etc., and usually ranged between 6 and 20 minutes (30 to 100 l of sampled air).

During the campaign a total of 42 flights were performed, which generated a total of 52 samples over 19 different days (Cruiser: 7 flights with a total of 17 samples over 6 days, Skywalker: 35 flights with a total of 35 samples over 16 days).

After the flights the samples were analyzed in the isostatic diffusion chamber FRIDGE. A sample substrate is placed on the cold table inside the sample cell. The evacuated cell is then inflated with water vapor. The combination of desired substrate temperature and ice supersaturation defines the exact pressure of water vapor that is inserted into the measurement cell. The water vapor rapidly activates the INP, and ice crystals begin to grow on the surface of these aerosol particles. A CCD camera monitors growth of ice (usually for 100 s) and a LabView controlled software automatically detects changes in the brightness



of the images of the emerging ice crystals. For this purpose the real time picture is compared to a reference picture taken prior to the measurement. After a measurement is completed the sample cell is evacuated again and the ice crystals evaporate completely. Then the temperature and relative humidity can be set to new conditions for the next activation.

During this campaign, samples were usually analyzed at $-20\,°C$, $-25\,°C$ and $-30\,°C$ and relative humidity of 95 %, 97 %, 99 % and 101 % with respect to water, or equivalently 115 to 135 % with respect to ice (see Tab. 1).

After the analysis, selected samples were examined by scanning electron microscopy (SEM) to gain information on the chemical composition and morphology.

For a detailed description of the sampling procedure and FRIDGE's measurement principle as well as its limitations and possible caveats, see Schrod et al. (2016).

## 2.5 Electron microscopy

Size and elemental composition of individual particles of selected samples were investigated by scanning electron microscopy using a field emission gun instrument (FEI ESEM Quanta 200 FEG, Eindhoven, The Netherlands), equipped with an energy-dispersive X-ray microanalysis system (EDX). Almost all ambient particle types are detectable by SEM/EDX analysis, but as SEM is a high vacuum method very volatile organic compounds (VVOC) will be lost. The samples were analyzed automatically by the software-controlled electron microscope (software EDAX/AMETEK GENESIS 5.231). Since the substrates cause a high silicon signal, silicon could not be used for classification of desert dust particles. Instead, aluminum was used for the identification of the Saharan components (in addition with: Mg, K, Ca, Ti and Fe). Besides the main alumosilicate group, Ti-rich alumosilicates and Ca-rich particles (either Ca (Mg) carbonates or mixtures of Ca carbonates with alumosilicates) were classified separately.

## 2.6 Measurements of aerosol number concentrations: Met One OPC

The Cruiser UAS was equipped with an optical particle counter (OPC, Met One Instruments, Model 212 Profiler) that measured the size distribution of airborne aerosol as a function of their optical diameter. The OPC reports aerosol particle number concentration with 1 Hz resolution in eight different channels ranging from 0.3 to 10 μm. The inlet of the OPC was preheated to keep relative humidity below 50 % to minimize the influence of water absorption onto particles.

## 2.7 Lidar

An automated multiwavelength PollyXT Raman polarization lidar with near-range capabilities (Althausen et al., 2009; Engelmann et al., 2016) was operated at Nicosia. This system emits linearly polarized light at 355, 532, and 1064 nm and has 10 receiver channels. The system is part of PollyNET (Baars et al., 2016) and was measuring around-the-clock autonomously. EARLINET quality standards were applied, e.g. the depolarization signal was calibrated automatically three times a day. The vertical and temporal resolution are 7.5 m and 30 s, respectively. More details can be found in Engelmann et al. (2016).



Products from the lidar used in this study are time-height-plots of the attenuated backscatter coefficient at 1064 nm and the volume depolarization ratio at 532 nm. All heights are above ground level. The attenuated backscatter coefficient is the calibrated range-corrected lidar signal (see e.g. Wandinger, 2012), i.e. it contains information on the backscattering by molecules and particles and is attenuated by extinction of these two types of scatterers. However, at 1064 nm, molecular scattering and

total extinction are very low, so that the attenuated backscatter coefficient at 1064 nm is nearly equal to the real particle backscatter coefficient. Thus, this product gives an indication about the amount of particles in the atmosphere. The lidar signal is calibrated using 2 hour mean profiles of extinction and backscatter coefficient obtained with the Raman method (Ansmann et al., 1992). The volume depolarization ratio is defined as the ratio of the backscattered light in orthogonal and parallel polarization plane with respect to the plane of polarization of the emitted light. It contains information from the whole volume, i.e.

molecules and particles. It is a measure of the non-sphericity of the observed scatterers, i.e. the higher the value the more non-spherical particles (e.g., dust) are present. At 532 nm Saharan dust typically yields a particle depolarization ratio of about 0.3, whereas spherical scatterers are considered to show a particle depolarization ratio of less than 0.05 (Mamouri and Ansmann, 2014, and references therein). Depolarization from molecular scattering is between 0.005 and 0.006.

Using the particle backscatter coefficient and the particle depolarization ratio at 532 nm, the number concentration of parti-

cles with a diameter > 0.5 μm can be estimated with an uncertainty of ± 30 % from lidar observations following the methodology of Mamouri and Ansmann (2016). A dust lidar ratio of 40 sr and a extinction-to-number conversion factor of 0.2 have been considered for that methodology according to the values provided by Mamouri and Ansmann (2016).

## 2.8  Dust transport model DREAM

In this study we used the Dust Regional Atmospheric Model - DREAM (Nickovic et al., 2001; Nickovic, 2005; Pejanovic et al.,

2011) driven by the National Centers for Environmental Predictions Nonhydrostatic Multiscale atmospheric Model - NMME (Janjic et al., 2001; Janjic, 2003; Janjic et al., 2011). The NMME-DREAM coupled modeling system has been developed to predict the atmospheric dust process, including dust emission from desert surfaces, horizontal and vertical turbulent mixing, long-range transport and deposition. It solves the Euler-type partial differential non-linear equation for dust mass continuity. Dust concentration is composed of eight bins with radii ranging from 0.15 to 7.1 μm. Dust emission in the model is proportional

to the intensity of the turbulent vertical mixing regimes (laminar, transient and turbulent mixing) near the surface. Specification of dust sources is based on the mapping of the areas that are dust-productive under favorable weather conditions. The USGS land cover data combined with the preferential sources of dust originating from the sediments in paleo-lake and riverine beds (Ginoux et al., 2001) have been used to define barren and arid soils as dust-productive areas. The North Africa - Middle East - Europe domain of the model has a horizontal resolution of 25 km; in the vertical, the model has 28 layers ranging from the

surface to 100 hPa. The initial and boundary atmospheric conditions for the NMME model have been updated every 24 hours using the ECMWF 0.5° analysis data. The concentration was set to zero at the "cold start" of DREAM, launched 4 days before April 1, thus permitting a 4-day spin up time to develop a meaningful concentration field at the date considered as an effective model start. After that time, 24-hour dust concentration forecasts from the previous-day runs have been declared as initial states for the next-day run of DREAM.





## 3 Results and Discussion

### 3.1 Mineral dust particles

During the entire month of the campaign, desert dust was frequently mobilized mainly from sources in the Central and Western Sahara and was injected into the prevailing westerly flow which carried it eastward across the Mediterranean Sea. Some of the

dust plumes were carried as far north as Central and Eastern Europe, but in general the filaments meandered eastward over the Mediterranean Sea and partly over Cyprus. In the atmosphere above Cyprus we encountered one major dust event ($Maj_{DE}$, April 8 to 11), two intermediate dust events ($Int_{DE1}$, April 15 to 16 and $Int_{DE2}$, April 21) and two minor dust events ($Min_{DE1}$, April 24 and $Min_{DE2}$, April 26) (Fig. 8). The classification of these different dust events builds on the peak dust concentrations in the operational layer of the UAS (0 to 3 km altitude) as predicted by the DREAM model: A dust event was classified as a)

major, when the model dust mass concentrations for more than 12 hours continuously exceeded 200 $\mu g\,m^{-3}$, b) as intermediate, between 100 and 200 $\mu g\,m^{-3}$ and c) as minor, between 50 and 100 $\mu g\,m^{-3}$. The corresponding time series of the predicted dust mass concentration is shown in Fig. 8a. Figure 8b shows lidar observations of the volume linear depolarization ratio at 532 nm for the same period. The agreement between the model and the observations is very good: a high dust concentration correlates to a high volume depolarization ratio ($R = 0.75$ for $n = 10850$ pairs of 3 h and 100 m interpolated data in the altitude range of

0 to 5 km agl).

The evolution of the dust load over the Mediterranean basin and its adjacent regions during the course of the campaign as predicted by the DREAM model is shown in the supplement (available online). Supplement S1 features a movie that depicts the dust load, i. e. the vertically integrated mass of mineral dust per surface area, over the stated regions (left panel). The movie starts on March 28 and ends on May 1 2016. The time step is 3 hours. The dust load ranges from 0.1 − 4 $g\,m^{-2}$, indicated by

colors from green over orange to purple. Blue contours show the 700 hPa geopotential height. The middle panel of the movie zooms in on Cyprus and its surrounding area. Here, the wind field at 3 km is indicated by arrows. The right panel displays the vertical distribution of dust above the UAS airfield site, using the same time step.

In the first days of the campaign (March 27 to April 7) no significant dust was observed throughout the lowest 10 kilometers of the atmosphere over Cyprus. On April 2 and 3 a very strong dust outbreak from sources in the western Sahara carried dust

far north across Europe, but did not affect the Eastern Mediterranean. On April 7 dust was mobilized from Central Saharan sources (ranging diagonally from northern Niger at 20° N, 10° E to northeastern Libya at 31° N, 23° E, dark brown ellipse in Fig 1b) and was advected by southwesterly flow directly towards Cyprus, causing the major dust event of the campaign on April 8 and 9. This plume impacted Cyprus with a layer of dust arriving at 3 to 4 km altitude above ground early on April 8. The concentration intensified until it peaked at 450 $\mu g\,m^{-3}$ at 2 to 3 km altitude in the night of April 8 to 9. The highest dust load

reached Cyprus at 6 UTC on April 9, when a homogeneous dust layer of 350 $\mu g\,m^{-3}$ between 1 and 6 km altitude swept across the island. Until 12 UTC DREAM dust concentrations still remained at the same high levels for a broad range of altitudes (1 − 4 km). Thereafter the concentration decreased steadily towards clean conditions, which prevailed after April 11. The next dust episode of intermediate strength was observed on April 15 and 16 with a peak of 150 $\mu g\,m^{-3}$ at about 2 − 4 km above ground level. For this case the DREAM model suggests source regions of the mineral dust in western and central North Africa (orange





rectangle in Fig 1b). After this dust layer passed over Cyprus, concentrations decreased while the dust sedimented on April 17. At about this date dust was mobilized in Central North Africa once more (yellow rectangle in Fig 1b). The corresponding intermediate dust event reached Cyprus on April 21 and showed peak concentrations above $150\,\mu g\,m^{-3}$ at 2 to 5 km altitude. In the last days of the campaign two minor dust plumes (April 24 and 26) traveled to Cyprus, both coming from sources in

northeastern Libya (light brown rectangle in Fig 1b).

## 3.2   Ice nucleating particles

The concentrations of INP measured from the UAS range over five orders of magnitude, when the full range of analyzing conditions is considered. Figure 9 shows the INP concentrations (color coded) as a function of relative humidity with respect to ice ($RH_{ice}$). At the lowest ice supersaturations and highest temperature tested (i.e. at $-20\,°C$), concentrations were typically

around $0.1\,\mathrm{std.l^{-1}}$ or below. Concentrations increased exponentially with ice supersaturation. At the highest relative humidity and lowest temperature (i.e. at $-30\,°C$) more than $100\,\mathrm{INP\,std.l^{-1}}$ were measured. Note that in sample 28, INP could not be analyzed quantitatively at the highest supersaturation and lowest temperature due to overloading, causing problems in the algorithm that distinguished and counted individual crystals. Samples 1 to 19, 21, 30 and 31 were only analyzed at $-25$ and $-30\,°C$. The INP concentrations increased from the beginning of the campaign until they reached a pronounced maximum on

April 9 ($Maj_{DE}$), followed by a minimum right after. The following days, INP concentrations were generally lower than in the first half of the campaign except for single events (e.g. April 21, $Int_{DE2}$). No flights could be performed between April 11 to 13 and April 16 to 20 due to unfavorable meteorological conditions and maintenance on the UAS.

The contribution of dust to local INP is reflected by the good agreement between INP and specific dust parameters, such as particulate matter (PM), aerosol optical thickness (AOT) and the modeled dust mass concentration. Figure 10 presents the

INP concentration at $T = -30\,°C$ and $RH_{ice} = 135.4\,\%$ (i.e. the top row of Fig. 9) from UAS together with coarse mode PM ($PM_{10}$-$PM_{2.5}$), AOT at 1020 nm measured at CAO and the dust mass concentrations calculated by the DREAM model for the sampling altitude. The mean UAS sampling altitude is given by the color coding of the symbols.

The mean vertical INP profile as derived from averaging INP concentrations in the height bins 0.5 to 1 km, 1 to 1.5 km, 1.5 to 2 km, 2 to 2.5 km is given in Fig. 11. The INP data in each altitude bin show considerable spread. However, the median

vertical profile shows the lowest INP concentration close to the surface and a gradual increase towards the top layer. The low INP concentration of around $1\,l^{-1}$ measured from FRIDGE samples at ground level at CAO is consistent with the observations made by the UAS. This reflects the overall situation during the campaign, in which layers of dust were frequently advected over the island at 2 to 3 km altitude. The lower parts of these layers were sampled by the UAS. The vertical profiles of dust calculated by the DREAM model (right panel of Supplement S1) support this view. The dominance of large scale dust advection can be

seen from the correlation between the levels of INP aloft and at the ground (Tab. 2). The highest correlation is found between INP and the total particle number concentration with diameters larger than $0.5\,\mu m$ ($n_{a>0.5}$) both measured on board the UAS ($R = 0.97$, $n = 11$). The correlation between the individual local concentrations of INP sampled from UAS and of the aerosol mass concentration calculated for the same sampling path by the DREAM model is $R = 0.69$, $n = 49$. INP from the UAS





are correlated to coarse mode PM ($R = 0.59$, $n = 49$) measured at CAO at ground level and to the vertically integrated AOT ($R = 0.31$, $n = 49$). Furthermore, the peaks of INP and the mineral dust parameters coincide (Fig. 10).

### 3.2.1 Parameterizations

The establishment of robust empirical correlations between ice nucleating properties and physical characteristics (size spectra) of atmospheric aerosol has been a challenge for decades (Georgii & Kleinjung, 1976), since it might allow to predict INP from aerosol data, which are much easier available than INP measurements. As presented above, we found that the INP concentration from UAS is highly correlated to the number of large particles measured simultaneously by OPC onboard. Thus, in the the following we will compare our INP measurements to recent particle-based empirical parameterizations of INP ($n_{\mathrm{INP}}(T_{\mathrm{k}})$). However, we have to bear in mind that the nucleation modes addressed by the different methods on which these parameterizations build do not perfectly overlap. From aircraft measurements in various different locations DeMott et al. (2010) derived the relationship (hereafter called D10):

$$n_{\mathrm{INP,\,D10}}(T_{\mathrm{k}}) = a(273.16 - T_{\mathrm{k}})^{b}(n_{a>0.5})^{(c(273.16-T_{\mathrm{k}})+d)}, \tag{1}$$

where the empirical parameters were set to $a = 0.0000594$, $b = 3.33$, $c = 0.0264$ and $d = 0.0033$. Equation 1 had been revised by DeMott et al. (2015) for mineral dust scenarios, i.e. the new parameterization (hereafter called D15) was obtained by compiling data from mineral dust INP collected in laboratory and from aircraft-based CFDC measurements inside mineral dust layers. D15 has come forward of the parameterizations of DeMott et al. (2010) and Tobo et al. (2013) and is given by Eq. 2:

$$n_{\mathrm{INP,\,D15}}(T_{\mathrm{k}}) = (cf)(n_{a>0.5})^{(\alpha(273.16-T_{\mathrm{k}})+\beta)} \exp(\gamma(273.16 - T_{\mathrm{k}}) + \delta), \tag{2}$$

The parameters $\alpha = 0$, $\beta = 1.25$, $\gamma = 0.46$ and $\delta = -11.6$ were empirically fitted by DeMott et al. (2015). The calibration factor $cf$ has been introduced to separately account for instrument-specific calibration and was set by default to $cf = 1$, but in special cases it shows a better fit when using $cf = 3$.

Figure 12 compares the INP concentration that we measured over Cyprus (for $RH_{\mathrm{water}}$ = 101 %, $T_{\mathrm{k}}$ = 253 K, 248 K and 243 K) to the INP concentration that is predicted for the same temperature (but somewhat higher RH) on the basis of the D10 and D15 parameterizations and the $n_{a>0.5}$ that was measured onboard the UAS. D10 is shown in Fig. 12a and D15 in Fig. 12b. The range of confidence intervals shown in red ($1\sigma$) and gray ($2\sigma$) are a little smaller for D10 than for D15. D10 predicts the high concentrations better than D15, but the lower concentrations are predicted poorly. The dust specific parameterization D15 on the other hand seems to be better suited to predict the observed measurements. While the slope of Fig. 12b is close to unity, the absolute values of the prediction are underestimated by one order of magnitude for these measurements. At this stage it remains difficult to determine whether this offset is caused by differences of a) instrumentation and measurement technique in general, b) the lacking overlap of freezing modes between the different experimental methods, c) different aerosol inlet systems (D15 used 1.5 to 2.4 µm cutoffs, whereas we used no cutoff), d) a variation in relative humidity (D15: 105 %, FRIDGE: 101 %), or a combination of all these. Furthermore not all of our samples may have been affected by dust. In a dust case study over Germany (Schrod et al., 2016) we have recently observed a slope close to unity between FRIDGE and D15, but with a 40 % underestimate ($T_{\mathrm{k}}$ = 257 K to 249 K) of the INP observed by FRIDGE as compared to the D15 parameterization.




However, in the case presented here, we do see a better fit between observed FRIDGE INP concentrations and D15 predicted immersion mode INP concentration when empirically setting $cf = 0.086$ (Fig. 12c), thus lowering the predicted values by a factor of 11. With this alteration the confidence intervals are narrowed markedly. Now 70 % of the data are within a factor of 3.28 and 97 % are within a factor of 14.37 around the 1:1 line. Even when allowing all the parameters to change freely we only

achieve a slightly better agreement (Fig. 12d). In this case the parameter $\gamma$ was set to $\gamma = 0.472$ while the $cf = 0.086$ and the other parameters were kept fixed. Then 70 % of the data are located within a factor of 2.63 and 97 % within a factor of 10.65 around the 1:1 line. Note that the variation of these constants (other than the prefactor $cf$) between this best fit case and the values given in D15 is very small.

### 3.2.2 Ice active fraction and active site density

The nucleating properties of the aerosol encountered over Cyprus may be characterized by its activated fraction ($AF$) as well as by the active site density ($n_\mathrm{s}$). Both parameters are a measure of how well the aerosol acted as a seed surface for ice nucleation. $AF$ stands for the fraction of INP out of the total aerosol particle number (Eq. 3), i.e. it indicates how many particles are needed in total to encounter one active ice nucleus:

$$AF = \frac{n_{\mathrm{INP}}}{(n_{a>0.5})} \tag{3}$$

The $n_\mathrm{s}$ parameter is an estimate of how many active sites are present upon the total aerosol surface ($s_{a>0.5}$) (Eq.4):

$$n_\mathrm{s} \approx \frac{n_{\mathrm{INP}}}{s_{a>0.5}} \tag{4}$$

Fig. 13 depicts $AF$ (a) and $n_\mathrm{s}$ (b) as functions of $RH_{\mathrm{ice}}$ in a box plot. Both parameters increase towards higher $RH_{\mathrm{ice}}$ in a similar manner, because they originate from the same aerosol and INP values. An exponential increase with $RH_{\mathrm{ice}}$ is discerned, which is characteristic for deposition freezing (Meyers et al., 1992). Some measurements are made nearly at the same

supersaturation, but at different temperatures (see the set of freezing conditions in Tab. 1). This yields an interesting result, that is somewhat difficult to interpret. It appears that in a sample that is analyzed at roughly the same $RH_{\mathrm{ice}}$ but at a different temperature, the higher INP counts are found at the higher temperature, which is opposite to the expectation. Since the number of measurements are relatively small, we cannot say with certainty whether this observation is a real effect or not. Possibly, the higher values at higher temperatures could be explained by condensation freezing starting to matter at RH close to water

saturation, whereas at the lower temperatures only the deposition mode is dominating the freezing process.

The values of $AF$ and $n_\mathrm{s}$ compare reasonably well with published measurements performed in regions influenced by mineral dust. E.g. Boose et al. (2016) found the active site density of two month measurement data in the summers of 2013 and 2014 at Izaña in Tenerife to range between $7 \times 10^7 - 3 \times 10^8$ m$^{-2}$ at $T = -25\,^\circ$C, $RH_{\mathrm{ice}} = 130$ % (this study: $2 \times 10^7 - 7 \times 10^8$ m$^{-2}$ at $T = -25\,^\circ$C, $RH_{\mathrm{ice}} = 129$ %) and between $2 \times 10^8 - 2 \times 10^9$ m$^{-2}$ at $T = -33\,^\circ$C, $RH_{\mathrm{ice}} = 135$ % (this study: $7 \times 10^7 - 2 \times 10^9$ m$^{-2}$

at $T = -30\,^\circ$C, $RH_{\mathrm{ice}} = 135$ %).

Furthermore, we did not find a significant variation of $AF$ and $n_\mathrm{s}$ during the campaign (not shown), indicating that the composition of the aerosol and its source might not have changed substantially between the different UAS flights. This suggests





that a background of mineral dust of fairly uniform nucleating properties was present over the Mediterranean Sea at all times
and it was only the burden of dust that affected the INP concentration that we measured.

### 3.3 Case study: Major dust event of April 9

On April 9 the highest concentration of both dust and INP were observed during the campaign (Fig. 10). Therefore, we present
5 an in-depth analysis of this day.

Figure 14 depicts vertical profiles of the four flights of that day together with the time series of the attenuated backscatter
and volume depolarization profile measured by lidar. The first two flights were performed with Cruiser, the last two with
Skywalker X8. Figure 14a shows the smoothed attenuated backscatter signal at 1064 nm of the lidar measurement in Nicosia,
30 km away from the UAS airfield. Most of the sampling altitudes (red) matched the layer of high backscattering (yellow
and red colors), containing mineral dust. The dust layer was located well above 1 km during the night and descended to
below 1 km in the morning hours, where it stayed after 9:00 UTC. Its vertical extent also shrank considerably between 4:00
and 9:00 UTC. The lower panel of Fig. 14 shows the volume depolarization ratio. A high depolarization ratio (yellow and
red colors) indicates the presence of non-spherical particles such as mineral dust. The concentration of INP (std.l$^{-1}$) along the
horizontal sampling tracks (red lines) are indicated by numbers (Fig. 14a). Most, but not all of the INP concentrations appear in
reasonable agreement with the lidar observations: high INP concentrations coincide with high backscatter coefficients and high
depolarization ratios. Note that sample 28 (second to last) showed such high ice activation even at $-25\,^{\circ}$C (180 INP std.l$^{-1}$),
that no measurement could be conducted at $-30\,^{\circ}$C. This is rather surprising since the sampling altitude was seemingly well
above the maximum of the actual dust layer (500 to 1000 m). The depolarization signal (Fig. 14b), on the other hand, showed
still a signal of medium strength for a broad range of altitudes up to 3 km, suggesting that a considerable amount of mineral
dust might have been collected. Furthermore, we cannot rule out that the observed differences might have been caused by a
heterogeneity in the dust spatial distribution between the two different operational sites.

The following paragraphs focus on the first Cruiser flight (red rectangle of Fig. 14, samples 24 and 25) of April 9. HYSPLIT
ensembles of backward air mass trajectories reaching the UAS airfield at elevation levels between 500 to 3000 m above ground
on April 9, 6 UTC are shown in Fig. 15. For every 500 m a new plot is shown (a to f). The ensemble option of the HYSPLIT
model computes each member of the ensemble by varying the initial meteorological starting condition of the model. Therefore,
this method can be used to estimate how likely it is that an air mass of a specific time and place was transported from a certain
region. Virtually all members of the ensemble of 27 trajectories ending at an elevation of 1500 m above ground or higher
(c to f) passed over the Sahara. Furthermore, the individual paths show very little spreading, suggesting that it is very likely
that the air masses traveled in the given direction. For 1000 m and especially 500 m, the trajectories diverged more strongly,
with only 63 % and 37 % of the trajectories passing over Northern Africa, respectively. This would suggest that sample 24
(05:30 to 05:41 UTC, mean alt. 1814 m agl) was very likely influenced by mineral dust transported from the Sahara and
sample 25 (05:48 to 05:58 UTC, 1006 m agl) was likely influenced by mineral dust transported from the Sahara.

These findings agree with the LIDAR measurements discussed above, and are supported by the high concentration of large
particles measured by OPC, which was the highest of all flights. The latter translates also into the highest INP concentrations





predicted (see Fig. 12), when the parameterizations are applied to these high $n_{a>0.5}$ values. The vertical profile and size spectra of $n_{a>0.5}$ during the flight are shown in Figs. 16 and 17. In Fig. 16 the three-dimensional flight track is plotted along with the color coded aerosol number concentration measured with the OPC. It illustrates a typical flight routine.

Right after the take-off the Cruiser UAS was set to ascent mode. While spiraling up the UAS accelerated up to its maximum
speed. The aerosol concentration for particles $d > 0.5\,\mu m$ increased from about $15\,cm^{-3}$ at ground level to $50\,cm^{-3}$ at $1\,km$ and remained more or less constant until Cruiser reached the maximum altitude (Fig. 17c). These measurements agree well with the aerosol particle concentration retrieved from lidar in Nicosia by the method introduced in Mamouri and Ansmann (2016) ($R = 0.96$ for $n = 19$ bins of $100\,m$ averages, OPC measurements: 05:05 to 05:55 UTC, lidar observation period: 06:50 to 07:00 UTC). Figure 17c also features the INP concentrations from the samples 24 and 25 and the INP concentration profile
based on $n_{a>0.5}$ from lidar and OPC observations and the D15 parametrization with best-fitting parameters from Fig. 12d. Figure 17d depicts the aerosol size distribution. Above $500\,m$ the number of particles as large as $2\,\mu m$ was clearly increasing, confirming the presence of a layer of mineral dust. Air temperature was stable at about 23 to $25\,°C$ in the lower $500\,m$ from where it decreased linearly with altitude at a rate of $-8\,°\,C\,km^{-1}$ (Fig. 17b). The relative humidity showed a dry layer between $0.2$ and $1\,km$ and a more humid layer above (Fig. 17a). As soon as the desired elevation was hit ($1800\,m$ agl), the electrostatic
sampling process started automatically for ten minutes. The UAS took an oval shaped course (Fig. 16), while maintaining the altitude until the sampling ended. After sampling was completed the UAS descended to the second pre-set height ($1000\,m$ agl) where the sampling process was repeated.

### 3.4 Electron microscopy of aerosol particles

After the ice nucleation analysis in FRIDGE two selected Si-wafers (samples No. 25 and 39) were analyzed with scanning
electron microscopy (SEM) equipped with an energy dispersive microanalysis system (EDX) for the elemental composition and morphology of the individual aerosol particles. Sample 25 was from the major dust event (April 9) and sample 39 from the second intermediate dust event (April 21). More than 1000 individual particles with diameters $> 400\,nm$ were analyzed (sample 25: 401 particles, sample 39: 628 particles). Overall, the chemical composition of the ambient aerosol sampled in the different dust events is very similar (Tab. 3, Fig. 18). For both samples around $99\,\%$ of the analyzed particles were determined
to be Saharan dust. In addition, a small number of Ca sulphates and carbonaceous particles was found. The Saharan dust particle category includes a main alumosilicate group, Ti-rich alumosilicates and Ca-rich particles, which are either Ca (Mg) carbonates or mixtures of Ca carbonates with alumosilicates. Furthermore, for both cases about $20\,\%$ carbonates were found, which are a good tracer for dust particles that have source regions in the North Sahara (Scheuvens et al., 2013, cf. section 3.1 and the supplement S1). From the present SEM analysis we cannot conclude on the chemical composition and nature of INP,
which make only a $10^{-3}$ to $10^{-5}$ fraction of the randomly selected particles on a wafer. Nevertheless, given the predominance of Saharan dust particles in those two samples, it seems likely that mineral dust particles also formed the majority of INP. Of course, it cannot be excluded that the ice-nucleating activity of the individual mineral dust particles will also be affected by minor or trace compounds present as thin surface coatings or small heterogeneous inclusions, which are hardly detectable in



SEM. For example, Conen et al. (2011) discussed that the carbon content/biological residues within dust samples can define their ice nucleation properties.

## 4 Summary and Conclusions

The atmosphere over Cyprus during the INUIT, BACCHUS and ACTRIS joint experiment was dominated by advection of dust in the lower and middle troposphere from North Africa, which coincided with high concentrations of INP. At ground level INP concentrations were an order of magnitude lower than aloft, pointing to relatively weak local marine and of terrestrial sources from Cyprus. From these pronounced vertical profiles we conclude that in atmospheric environments that are affected by the large dust sources of the globe, INP measurements performed at ground level will be only of limited significance for the situation several kilometers aloft at cloud level, for which this information is needed. Although the situation encountered by us over the Mediterranean Sea must not be generalized, it is well known as a climatological feature that desert dust regularly travels over distances of thousands of kilometers in this altitude range (Prospero, 1999; Liu et al., 2008).

Several events of long-range transport of Saharan mineral dust with varying intensity were registered during the campaign. The INP concentration followed various dust proxy parameters, and correlated well with the number of large aerosol particles ($d > 0.5\,\mu m$) measured in-flight as well as with the dust mass modeled by DREAM. The ice-active site density of the aerosol encountered during the flights compared reasonably well with published data from the Sahara (Boose et al., 2016). SEM analysis of samples taken during the two strongest dust events of the campaign identified about 99 % of the individual aerosol particles with diameters above 400 nm as dust from the North Sahara. Concomitant with the strongest dust event, the INP concentrations reached a peak value of more than 100 active ice nuclei per liter air. The measurements allow no conclusion whether it is dust that actively nucleates ice or particles that are admixed and travel with the dust.

These are the first INP measurements obtained with the technology of unmanned aircraft systems. The combination of an UAS and an offline sampling system with subsequent laboratory analysis of INP is novel and represents a promising alternative to measurements on a research aircraft. Aside from the simplicity as compared to a conventional aircraft mission, the main advantage of the combination of UAS and INP sampling device is its versatility. We were able to adapt rapidly to the current meteorological situation, thereby scheduling flights to accurately target specific small-to-medium scale phenomena such as dense layers of dust. However, admittedly, the small to medium sized UAS have certain limitations in terms of maximum payload, top elevation, flight time, spatial coverage and meteorological conditions (wind speed / precipitation), as well as flight restrictions due to safety of air traffic.

Nevertheless, we encourage other groups to consider UAS as an option to carry out measurements of ice nucleating particles, whether with a similar setup as presented here or in a any different configuration. This tool could broaden the few existing sets of non-surface based INP observations significantly for regions all over the world.

*Acknowledgements.* The research leading to these results has received funding from the European Union's Seventh Framework Programme (FP7/2007-2013) project BACCHUS under grant agreement No 603445 and the Deutsche Forschungsgemeinschaft (DFG) under the Re-





search Unit FOR 1525 (INUIT). This campaign has been performed at the Cyprus Atmospheric Observatory (CAO) which is part of the ACTRIS2 project that has received funding from the European Union's Horizon 2020 research and innovation programme under grant agreement No 654109. The support by the international Research Institute for Climate and Society, Columbia University, Palisades, N.Y. with meteorological data and software is gratefully acknowledged. We acknowledge support from the DFG-Research Center / Cluster of Excellence "The Ocean in the Earth System-MARUM". We thank SNO PHOTONS/AERONET from INSU/CNRS/University of Lille, France, AERONET-Europe/ACTRIS Calibration Center and AERONET Team at GSFC for their kind cooperation. The Department of Labour Inspection (DLI, Ministry of Labour, Welfare, and Social Insurance) is thanked for the provisions of ground-based weather and PM data at CAO. The provision of the HYSPLIT transport and dispersion model from the NOAA Air Resources Laboratory is gratefully acknowledged.



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





**Table 1.** Thermodynamic conditions of INP analysis in FRIDGE.

| $T$ [$^{\circ}$C] | $RH_{\text{water}}$ [%] | $RH_{\text{ice}}$ [%] |
|---|---|---|
| -20 | 95 | 115.6 |
|  | 97 | 118.0 |
|  | 99 | 120.4 |
|  | 101 | 122.9 |
| -25 | 95 | 121.3 |
|  | 97 | 123.9 |
|  | 99 | 126.4 |
|  | 101 | 129.0 |
| -30 | 95 | 127.4 |
|  | 97 | 130.1 |
|  | 99 | 132.7 |
|  | 101 | 135.4 |

**Table 2.** Correlation coefficients of INP concentration sampled from UAS ($T = -30\,^{\circ}$C, $RH_{\text{ice}} = 135.4$ %) to dust related parameters.

| dust parameter | $R$ | $n$ | platform / location |
|---|---|---|---|
| $n_{a>0.5\,\mu\text{m}}$ | 0.97 | 11 | UAS path |
| DREAM dust mass concentration | 0.69 | 49 | UAS path |
| PM | 0.59 | 49 | ground, at CAO |
| AOT | 0.31 | 49 | total column above CAO |





**Table 3.** Average chemical composition of randomly selected single aerosol particles analyzed by electron microscopy of Sample 25 (April 9) and Sample 39 (April 21).

|  | Sample 25 count | Sample 39 count |
|---|---|---|
| total particles analyzed | 401 | 628 |
| Saharan dust particles | 395 | 620 |
| carbonaceous particles | 4 | 8 |
| other | 2 | 0 |
| Saharan dust particles only |  |  |
| alumosilicates (Mg, K, Fe) | 283 (72 %) | 463 (75 %) |
| alumosilicates (Ti-rich) | 23 (6 %) | 21 (3 %) |
| Ca-rich alumosilicates / Ca (Mg) carbonates | 84 (21 %) | 130 (21 %) |
| gypsum | 5 (1 %) | 6 (1 %) |



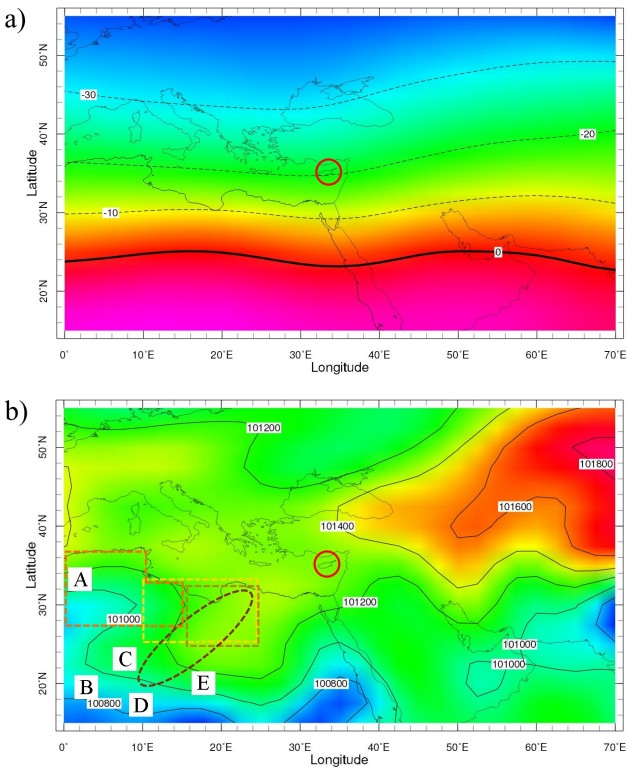

**Figure 1.** a) Mean stream function at $500\,\mathrm{hPa}$ (range: $-40$ (blue) to 10 (purple) $\mathrm{m^2 s^{-1}}$). b) mean sea level pressure (b, range: 100600 (blue) to 102000 (purple) $\mathrm{Pa}$) in April 2016 for the Eastern Mediterranean (IRI 2016). The island of Cyprus is indicated by a red circle. Letters A to E in the lower panel refer to North African orography: A = Foothills of the Atlas mountains, B = Adrar Plateau, C = Ahaggar mountains, D = Air massif and E = Tibesti mountains. The dashed rectangles and ellipse refer to dust source regions for this campaign (cf. section 3.1 and Supplement S1)





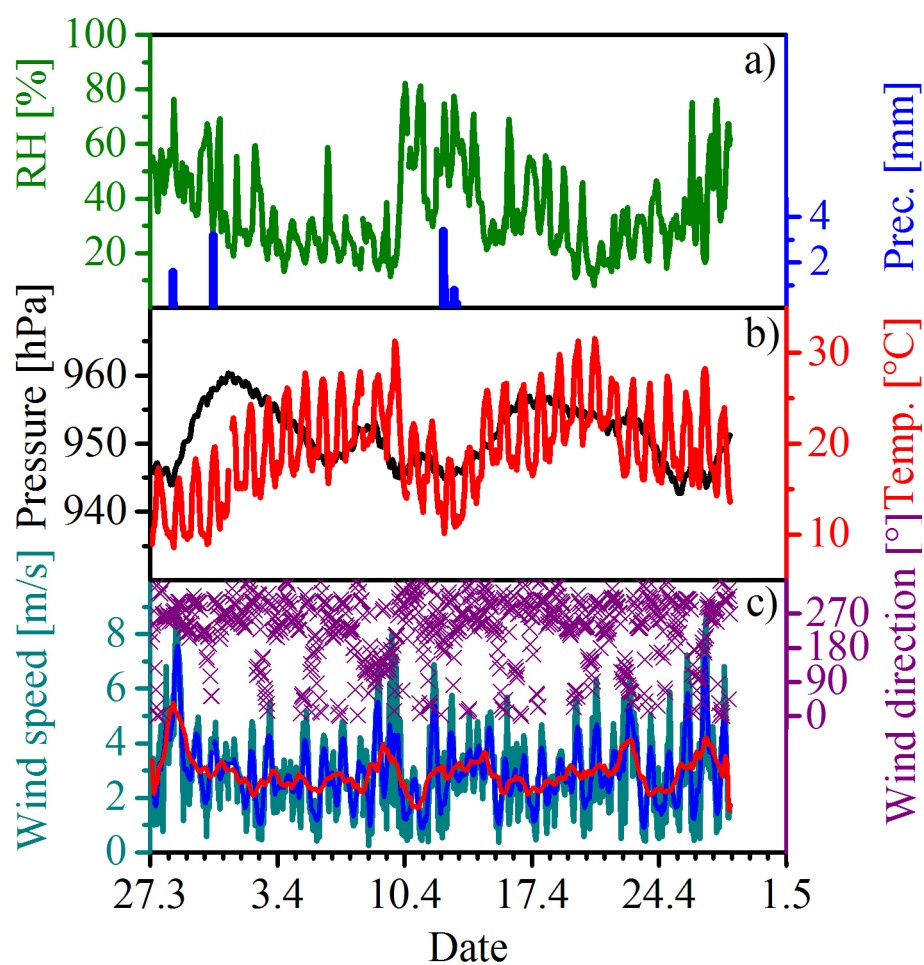

**Figure 2.** 1-h averages of meteorological parameters (a: relative humidity (green) and precipitation (blue), b: unreduced pressure (black) and temperature (red), c: wind direction (purple crosses) and speed (cyan) and its running means of 6 hours (blue) and 24 hours (red)) during the campaign measured at CAO by the Department of Labour Inspection of Cyprus (DLI).




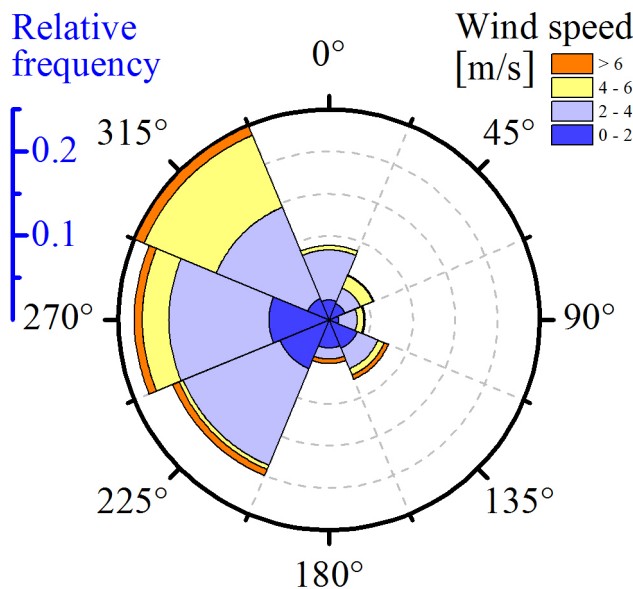

**Figure 3.** Wind rose based on hourly averages of wind speed and direction measured at CAO by DLI.

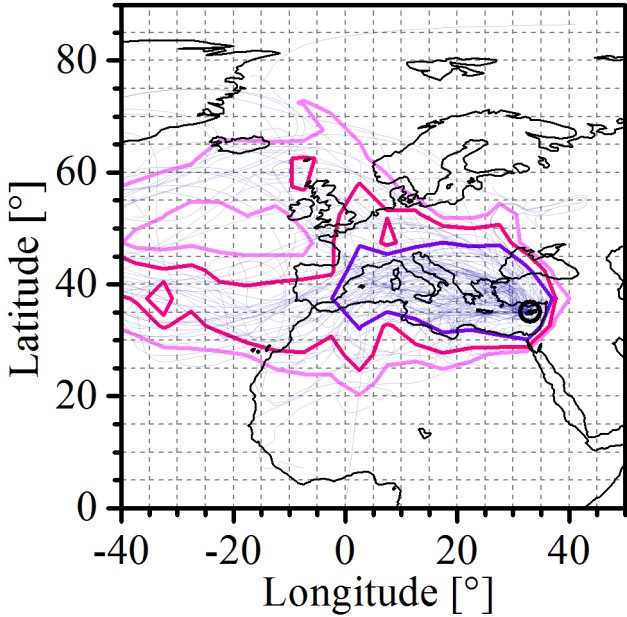

**Figure 4.** Frequency of trajectories arriving at the UAS airfield (black circle) during the campaign. Back trajectories (10 days) were computed with HYSPLIT (NOAA-ARL, GDAS1, start height 1000 m, Stein et al., 2015; Rolph, 2016). Three back trajectories (faint blue lines) were initiated for each day of the campaign (03:00, 06:00, 09:00 UTC). More than 5 % of the trajectories touch the area that lies inside the light pink lines, more than 10 % inside the dark pink lines and more than 20 % inside the purple line (based on a 5° by 5° grid size).



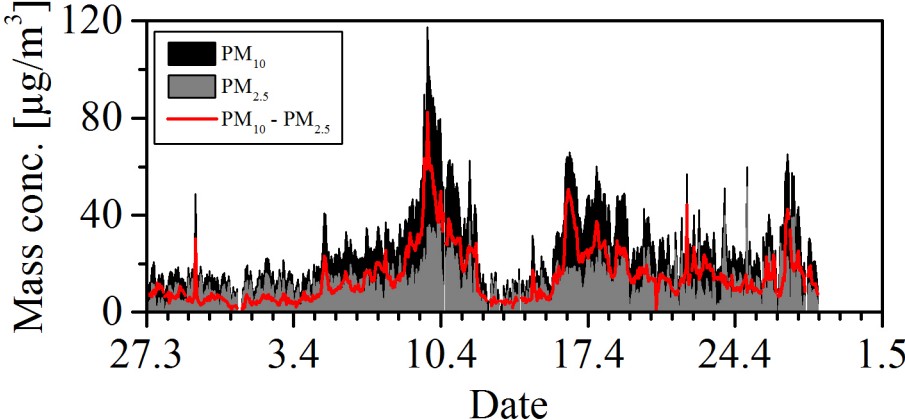

**Figure 5.** Aerosol mass concentrations PM$_{10}$ (black), PM$_{2.5}$ (grey), and the difference between both, i.e. the coarse mode (red) measured at CAO by DLI.

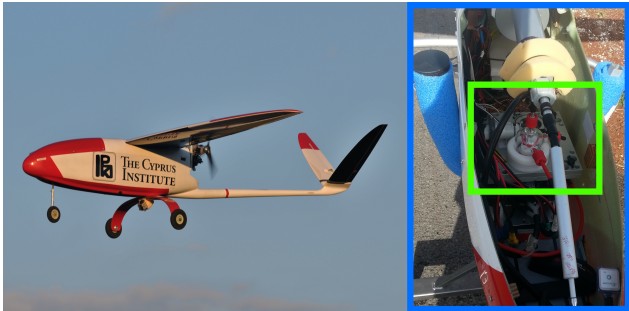

**Figure 6.** The Cruiser UAS with the multi-sample aerosol collector (green rectangle) (photograph by: Kjell-Sture Johansen).

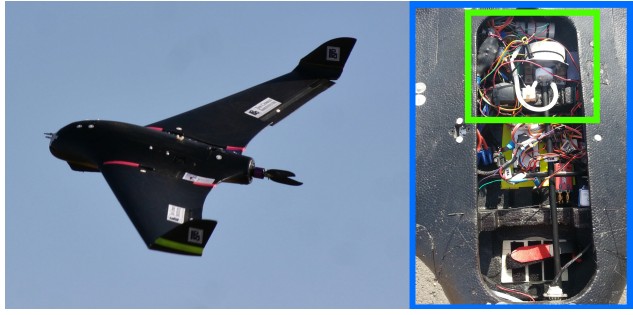

**Figure 7.** The Skywalker UAS with the single-sample aerosol collector (green rectangle) (photograph by: Christos Keleshis).

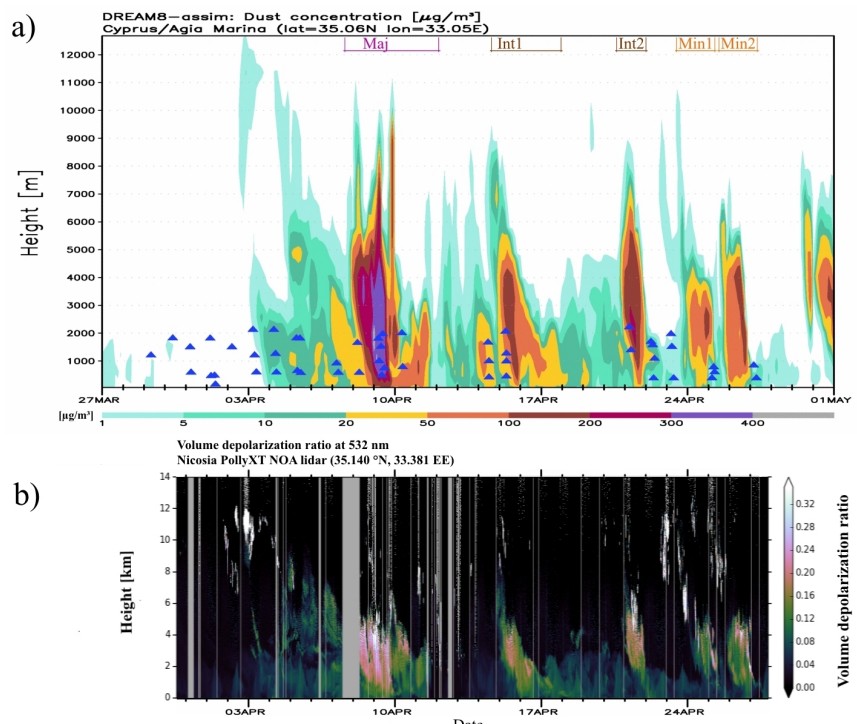

**Figure 8.** a) Time series of dust mass concentration vs. altitude predicted by the DREAM model for the UAS airfield site. Episodes of mineral dust are categorized as major ($Maj_{DE}$), intermediate ($Int_{DE1}$, $Int_{DE2}$) or minor ($Min_{DE1}$, $Min_{DE2}$) dust events, see text. Blue triangles in label indicate UAS flights with INP sampling. b) Time series of volume depolarization ratio at 532 nm vs. height obtained from PollyXT Lidar in Nicosia.




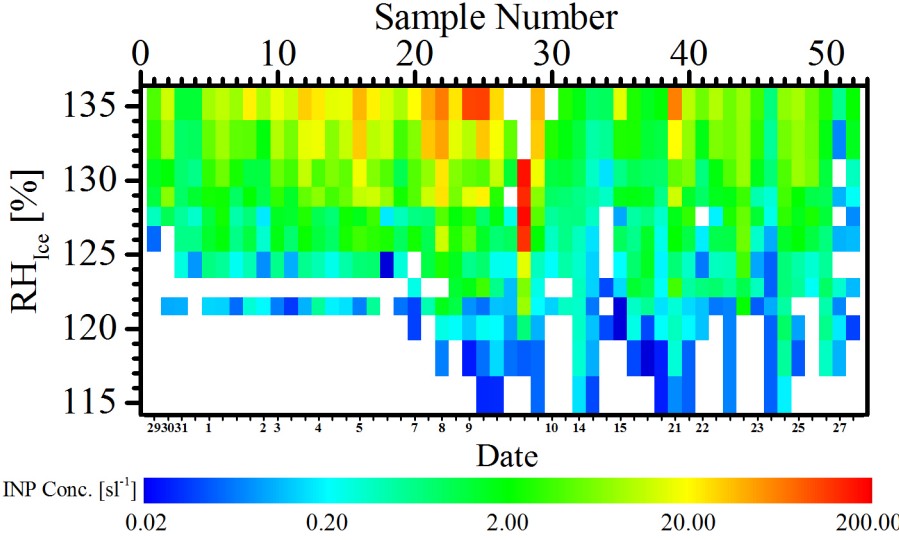

**Figure 9.** INP concentrations (color coded) as function of relative humidity with respect to ice in the temperature range from $-20$ to $-30\,°C$ (see Tab. 1) observed during all UAS flights. The date (in days of March (29 to 31) and April (1 to 27)) corresponding to the sample number is given on the lower abscissa. Where no number appears, the date is the same as the previous sample(s).

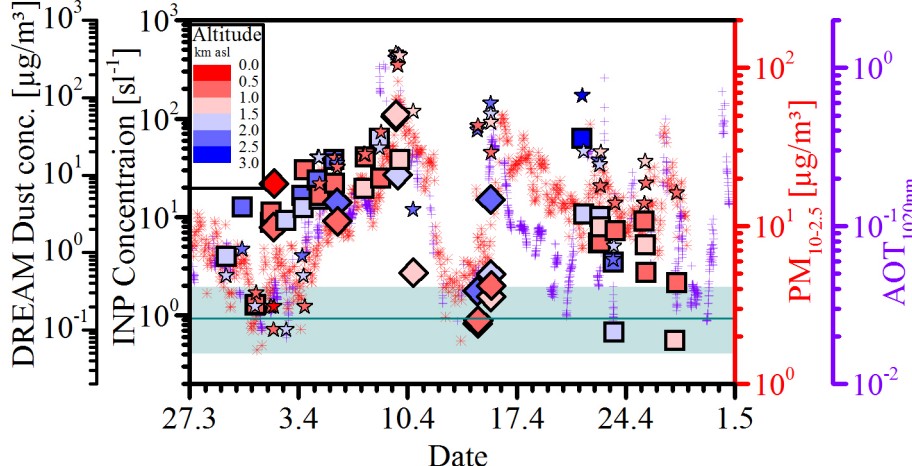

**Figure 10.** INP concentration ($T = -30\,°C$, $RH_{ice} = 135.4\,\%$) sampled during flights with Skywalker (squares) and Cruiser (diamonds). The symbols are color-coded by the mean sampling altitude. The cyan line gives the median INP concentration of 33 FRIDGE measurements from this campaign at ground level at CAO (532 m asl), the shaded area shows the interquartile range. The stars represent the DREAM model output of the dust mass concentration. The dust proxies PM and AOT (Goloumb, personal communication) are indicated in red and purple.



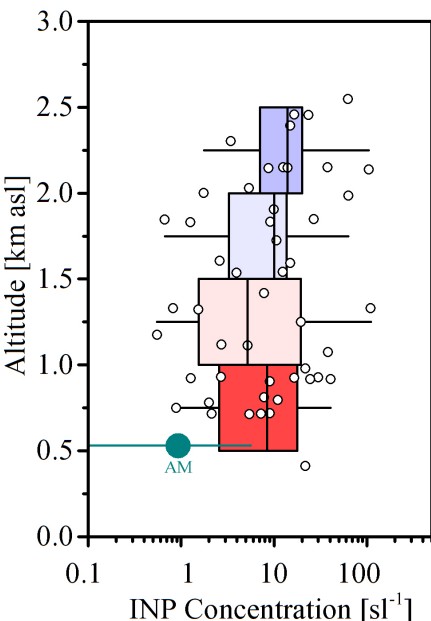

**Figure 11.** INP concentrations ($T = -30\,°C$, $RH_{ice} = 135.4\,\%$, circles) versus sampling altitude. The boxes (colors same as Fig. 10) show the interquartile range of the data with the median given as a vertical solid line, the black bars give the full range of concentrations. The median INP concentration of 33 FRIDGE measurements from this campaign at CAO in Agia Marina (AM, 532 m asl) is indicated by a the cyan circle, the range is shown by a cyan bar.




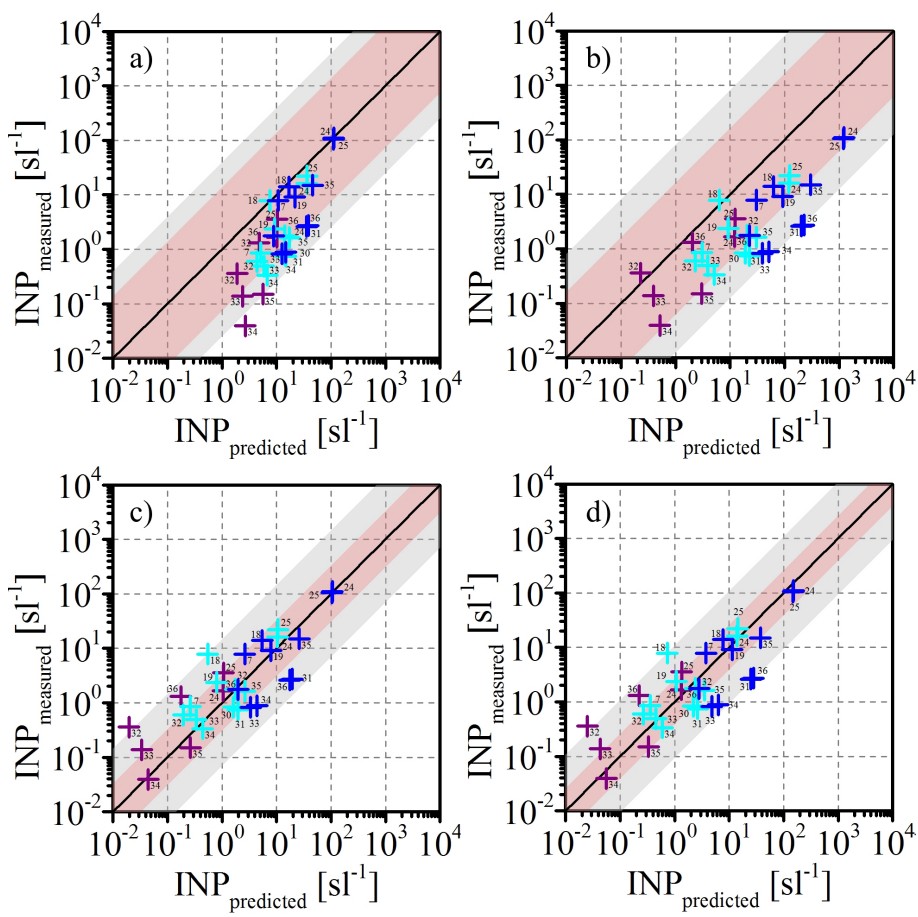

**Figure 12.** Scatter plot of INP concentration measured by FRIDGE against the same as predicted by a) D10 and b) D15. Label c) uses the same parameterization as b) with the exception that $cf$ is set to the best fit of 0.086. Label d) also has the same fixed values for $\alpha$, $\beta$, $\delta$ and $cf$ as in c) but the constant $\gamma$ is fitted to best represent the measured data. The confidence intervals of $1\sigma$ (70%) and $2\sigma$ (97%) are indicated by a red and grey belt around the 1:1 line. FRIDGE data were measured at $RH_\text{water}$ = 101 % and temperatures $-20\,^\circ$C (purple), $-25\,^\circ$C (cyan) and $-30\,^\circ$C (blue) and had valid OPC measurements carried out simultaneously aboard of Cruiser. Numbers beside the symbols corresponds to the sample numbers from Fig. 9.



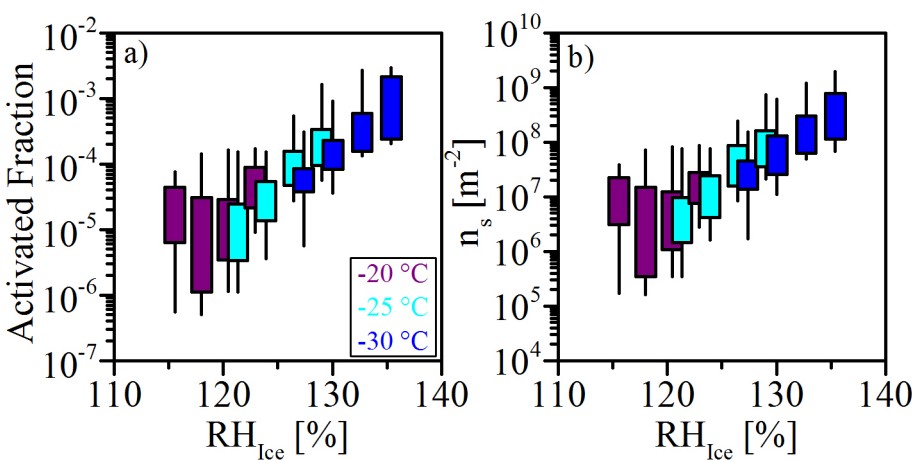

**Figure 13.** Activated Fraction $AF$ (a) and active site density $n_s$ (b) of aerosol particles larger than 0.5 µm in diameter as a function of ice supersaturation for Cruiser flights. The boxes represent the interquartile range. Vertical lines give the full range of observations. The width of the boxes represents the uncertainty in humidity ($\pm\,1\,\%$).





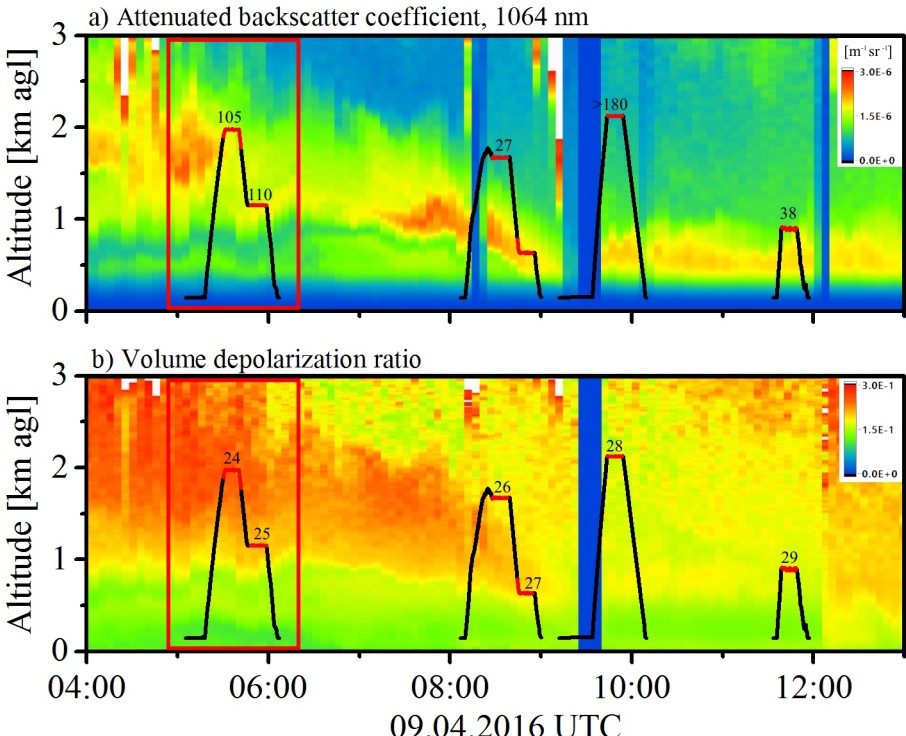

**Figure 14.** Time series of lidar observations measured in Nicosia on April 9. The upper panel shows the attenuated backscatter signal at 1064 nm and the lower panel the volume depolarization ratio. The smoothed lidar image is superimposed with the UAS flight track (black). Periods of sampling are indicated in red (numbers 24 to 29 in label b). The numbers above the sampling period in label a) give the INP concentration in $\mathrm{std.l^{-1}}$ at $-30\,^{\circ}\mathrm{C}$ and $135.4\,\%$ ice saturation. Altitude is given in meters above ground level relative to the location of the lidar in Nicosia (180 m asl). The red rectangle refers to the time periods investigated in Figs. 15, 16 and 17.

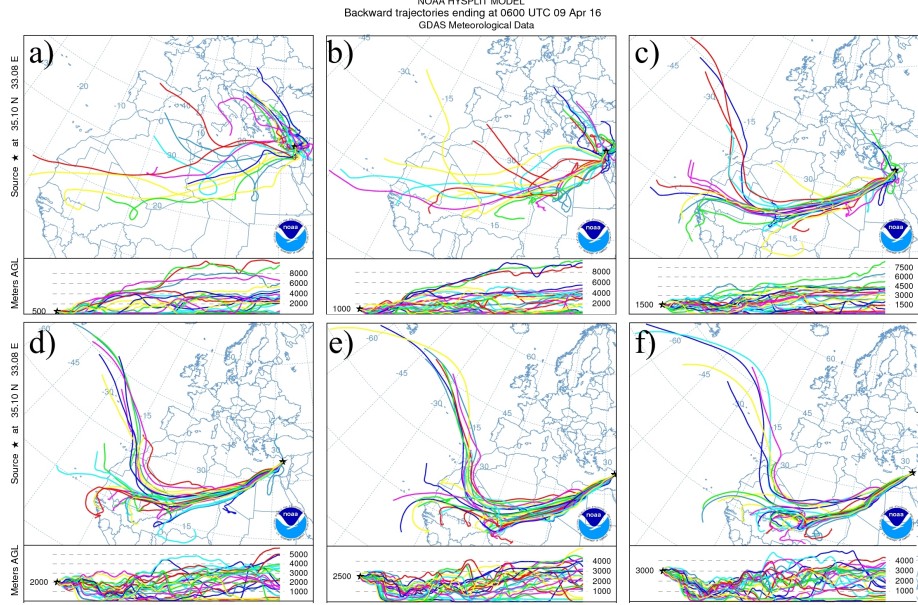

**Figure 15.** Ensembles of 27 seven day HYSPLIT backwards trajectories (Stein et al., 2015; Rolph, 2016) for the starting heights of 500 (a), 1000 (b), 1500 (c), 2000 (d), 2500 (e) and 3000 (f) m agl calculated for the UAS airfield on April 9, 6 UTC.

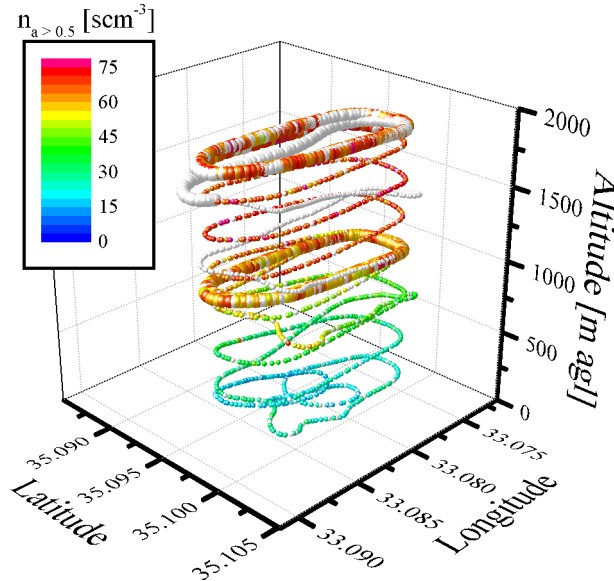

**Figure 16.** Flight track of the first Cruiser flight on April 9. Colored spheres give the total number concentration of particles larger than 0.5 μm (white: no measurement available). Larger sized spheres indicate when sampling was carried out.



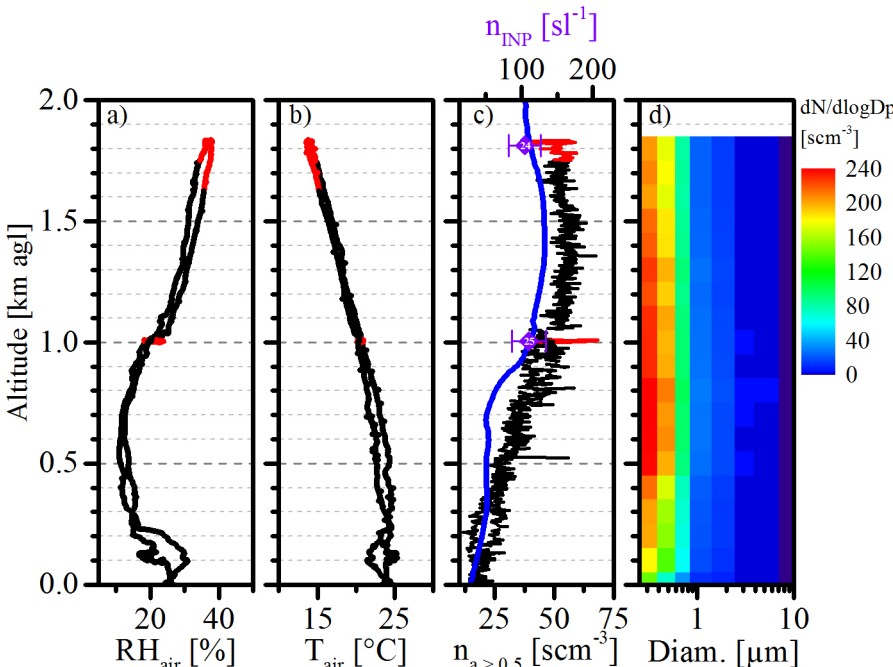

**Figure 17.** Vertical profiles of the first Cruiser flights on April 9: a) relative humidity, b) temperature, c) (lower abscissa) total number concentration of particles with a diameter larger than 0.5 µm measured by OPC on board the UAS (black) and retrieved from lidar observations (blue) and c) (upper abscissa) the corresponding INP concentration at $T = -30\,°C$ from OPC and lidar as predicted by the parametrization shown in Fig. 12d. The INP measurements of the samples 24 and 25 are given by purple diamonds. Altitude is given in meters above ground level relative to the location of the UAS airfield (327 m asl). The INP sampling interval is indicated in red. Panel d) shows the average aerosol size distribution is binned into 100 m intervals.





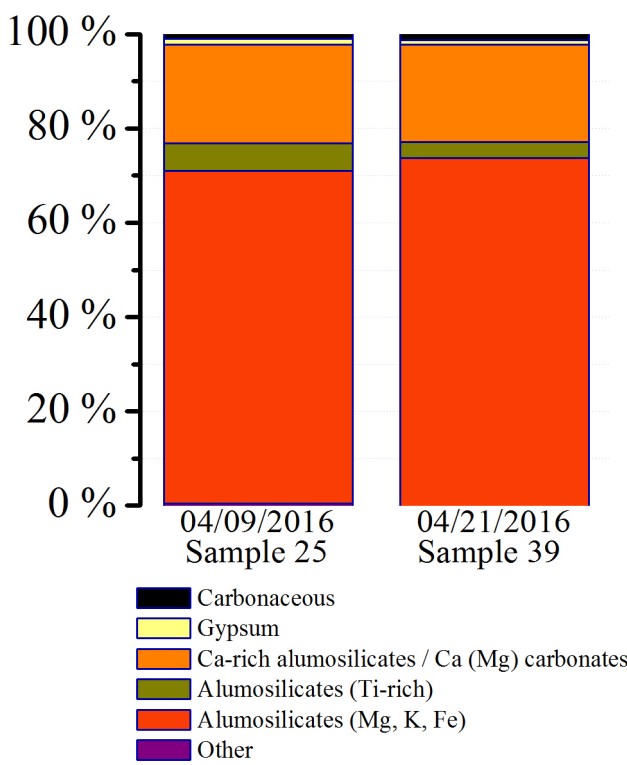

**Figure 18.** Chemical composition of single aerosol particles from electron microscopy of Sample 25 (April 9) and Sample 39 (April 21).