# Peer review of "Ice nucleating particles over the Eastern Mediterranean measured by unmanned aircraft systems"

_Atmospheric Chemistry and Physics, 2016_

## Referee Comment (RC1) · Anonymous Referee #1 · 11 Jan 2017

Review of "Ice nucleating particles of the Eastern Mediterranean measured by unmanned aircraft systems" by Schrod et al.

In this paper, Schrod et al. leverage the use unmanned aircraft systems to measure the abundance of ice nucleating particles which are active in the immersion and condensation modes in the lower troposphere for the first time. The study is conducted in a region frequently influenced by Saharan dust emissions, making the results of particular importance to furthering our understandings of how desert dusts can impact upon clouds. During the study, a number of long-range transported dust events are captured, and the importance of dust as an INP in this environment is highlighted by correlation to PM10 mass, aerosol optical thickness and modelled dust concentrations.

The paper itself is well-written, and the work presented is both novel and likely to be of large interest to researchers interested in this topic. Even of the few remarks I have on the manuscript, most of these are relatively minor. As such, I recommend the paper for publication after consideration of the following:

Comments:

P1L5: It should be noted that here, and during all the other occurrences throughout the paper, that the plural of ice nucleating particle is "INPs" and not "INP"

P6L18: What is the efficiency of the sampling system (i.e. inlet + aerosol sampling unit) for different particle sizes?

P7L8-9: I think a very brief discussion of the limitations and possible caveats of the FRIDGEs measurement principles is pertinent here. While these are listed in Schrod (2016), a brief summary here would also be useful, as these are of course also central to this work.

P11L19: A brief mention as to what the physical meaning of the calibration factor, cf, is would be useful to the reader here.

P12L26-29: Can you show this good agreement by using a plot?

Section 3.2.2.: As you have ns values, it might be interesting to compare your results to the lab-based parameterisation for the ice nucleating activity of mineral dusts developed by Niemand et al. (2012) and maybe even to that for feldspar by Atkinson et al. (2013), taking into account that only a fraction of the dusts are likely to be feldspar

P15, Line1: It should be noted here that Conen et al. examined the ice nucleating activities of dusts at much warmer temperatures than were probed here. That being said, a similar point is made by Tobo et al. (2014), which might be good to reference here.

References:

Atkinson, J. D., Murray, B. J., Woodhouse, M. T., Whale, T. F., Baustian, K. J., Carslaw, K. S., Dobbie, S., O'Sullivan, D. and Malkin, T. L.: The importance of feldspar for ice nucleation by mineral dust in mixed-phase clouds., Nature, 498(7454), 355–8, doi:10.1038/nature12278, 2013.

Niemand, M., Möhler, O., Vogel, B., Vogel, H., Hoose, C., Connolly, P., Klein, H., Bingemer, H.,

DeMott, P. and Skrotzki, J.: A particle-surface-area-based parameterization of immersion freezing on desert dust particles, J. Atmos. Sci., (2012), 2012.

Tobo, Y., DeMott, P. J., Hill, T. C. J., Prenni, A. J., Swoboda-Colberg, N. G., Franc, G. D. and Kreidenweis, S. M.: Organic matter matters for ice nuclei of agricultural soil origin, Atmos. Chem. Phys., 14(16), 8521–8531, doi:10.5194/acp-14-8521-2014, 2014.

---

## Referee Comment (RC2) · Anonymous Referee #2 · 20 Jan 2017

This manuscript reports on a new technique, determining ice nucleating particle concentrations at various altitudes from drones. The technique clearly shows promise, and the measurements are needed in this area. I have a few questions/concerns, which the authors need to address prior to publication. Then, the manuscript will make a solid contribution to the field.

Major Comments: 1. The abstract ends with a bold and interesting conclusion, that ground level INP measurements are of limited use in understanding INP aloft and their role in cloud formation. This may be true, but it is not substantiated by the data as presented. More could be done with the collected data set, as I describe below.

2. Pg. 5. The cruiser has a 2-stroke engine. 2 stroke engines are notoriously dirty running, producing significant pollutants, including NOX and likely particulates as well.

[Figure]

There is a serious concern that the emissions from the engine will be active INP and thus will contaminate the INP sample and bias the reported concentrations. What has been done to check, correct and/or avoid this?

3. pg. 6 ln 27. Sampling times and therefore volumes vary by a factor of 3. Why wasn't the sample time kept uniform? What impact does this have on INP concentration results? This needs to be addressed.

4. pg 10, ln 30- pg 11 ln 7 and Figures 11&12: The manuscript states the INP concentration is highly correlated to the concentration of large particle measured by the OPC and with the vertically integrated aerosol optical depth. Unfortunately, not all the data presented supports this conclusion. - This is the major issue with the manuscript.

In Figure 11, we see that INP correlations were essentially unchanged over the vertical altitudes sampled. (Particle concentrations are not included in the figure, but it is highly unlikely that they are equally invariable). When all the data is combined into one plot, interesting details are often lost, and I suspect that that is the problem here. Aren't there cases in which a dust layer aloft was sampled? Did INP concentrations increase with the dust layer? Yes or no? Perhaps dust layers were too thin to have an impact on the concentration of INP in any given sample? Or, did the drone miss the layers? Alternatively, what about contamination from the drone exhaust? Is that somehow causing falsely high values at lower altitudes (i.e. take-off and lands)

Rather than look at overall campaign correlations (i.e. Figure 12), it should be much more telling to look at cases. Likewise, correlations with particle counts are expected to vary vertically, especially when dust layers aloft are an order of magnitude higher than at ground level. The lidar is evidence that that as occured in some cases here, but the connection to INP concentration is not demonstrated.

In Figure 14, where a case study is presented, peaks in INP concentrations do not coincide with peaks in backscatter coefficient, especially the point at 10:00 UTC of 180 INP/std l. Any idea what happened at this point.

Backscattering coefficient can be complicated by many particle characteristics. - It would be nice to also include the OPC concentrations along with the INP concentrations. Also, in 14C depolarization ration and INP are not correlated in any way.

In summary, the vertical profile of INP may be contaminated by the drone's engine exhaust? And INP concentrations may or may not be sensitive to cases of dust events. Given the wealth of data collected here including all the key elements to really look at dust, dust size, and INP at multiple altitudes, I urge the authors to consider additional cases are available which support their statement that INP is highly correlated to large particles. If the cases do not support that correlation, other interpretations of the data should be considered.

5. On figure 12, it is impressive that the predicted numbers are lower than measurements in 12a and 12b, but well correlated in 12c&d. It would be nice to expand the discussion of how these were parameterized differently, as it seems that this result is a key finding of the manuscript.
* * *

---

## Author Comment (AC1) · 28 Feb 2017

**Response to Anonymous Referee #1**

First of all, we want to thank the referee for submitting his/her helpful and productive annotations, which led to improvements and clarifications within the manuscript.

We have prepared a revised manuscript that addresses the questions and comments of the referees. Furthermore, below we explicitly respond to each of the items raised in the comments of anonymous referee #1. These comments are indicated by using *italics,* whereas the author's response is presented in blue. Changes in the manuscript are given in green; changes to the supplement are given in purple. The differences are also highlighted in separate PDFs using latexdiff. All line and page numbers refer to the ACPD manuscript version, not the revised manuscript.
* * *
**Review of "Ice nucleating particles of the Eastern Mediterranean measured by unmanned aircraft systems" by Schrod et al.**

*In this paper, Schrod et al. leverage the use unmanned aircraft systems to measure the abundance of ice nucleating particles which are active in the immersion and condensation modes in the lower troposphere for the first time. The study is conducted in a region frequently influenced by Saharan dust emissions, making the results of particular importance to furthering our understandings of how desert dusts can impact upon clouds. During the study, a number of long-range transported dust events are captured, and the importance of dust as an INP in this environment is highlighted by correlation to PM10 mass, aerosol optical thickness and modelled dust concentrations.*
*The paper itself is well-written, and the work presented is both novel and likely to be of large interest to researchers interested in this topic. Even of the few remarks I have on the manuscript, most of these are relatively minor. As such, I recommend the paper for publication after consideration of the following:*

**Comments:**

*Pg. 1, ln. 5: It should be noted that here, and during all the other occurrences throughout the paper, that the plural of ice nucleating particle is "INPs" and not "INP"*
We thank the referee for noticing and have implemented the plural as suggested.

*Pg. 6, ln. 18: What is the efficiency of the sampling system (i.e. inlet + aerosol sampling unit) for different particle sizes?*
We will answer this comment below, add this discussion to the supplement and add a remark to the manuscript.

Pg.6, ln.18 now reads:

Both UAS were equipped with a customized inlet  nozzle that was connected by tubing to  the aerosol sampling unit. The diameter $D_s$ of the sample inlet nozzle was such that near isokinetic sampling was achieved at the average air speed $U_0$ of the UAS and sampling rate $Q$. The error due to anisokinetic sampling was estimated to be

typically less than 20% of particle number for particles up to 10 µm in diameter. A detailed discussion of sampling errors due to anisokinetic sampling is presented in the supplement.

The following paragraph was added to the supplement:

**2  Estimate of errors from anisokinetic sampling**

The diameter $D_s$ of the sample inlet nozzle was such that near isokinetic sampling (i.e. the air sample inlet velocity $U$ equals the air speed $U_0$ of the UAS) was achieved for the average operational air speed $U_0$ of the UAS and the mean aerosol sampling rate $Q$. The three quantities $U_0$, $D_s$ and $Q$ are related by Eq. 1:

$$D_s = 2\sqrt{\frac{Q}{\pi U_0}} \tag{1}$$

For the mean operational conditions these parameters are:   a) for Cruiser: $U_0 = 27.8$ m s$^{-1}$, $Q = 5$ lpm, $D_s = 1.95$ mm; and b) for Skywalker: $U_0 = 16.7$ m s$^{-1}$, $Q = 5$ lpm, $D_s = 2.52$ mm.

In the following we estimate sampling errors due to anisokinetic conditions (i.e. $U \neq U_0$). The latter may arise when the UAS spirals in the wind field and air speed $U_0$ and pump rate $Q$ deviate from conditions a) or b) due to tail wind or head wind. The range of $U_0$ and $Q$ observed during the campaign is given in Tab. S1. All parameters vary typically by less than 20%. Our estimate follows the discussion of sampling errors in Hinds (1999), chapter 10. For simplicity's sake and for the lack of other measurements, we will use the assumption that the gas streamlines entering the sampling inlet show no misalignment whatsoever. For this idealized case the ratio of the aerosol number concentration $C$ at $U \neq U_0$ to $C_0$ at isokinetic conditions is then given by Belyaev and Levin (1974):

$$\frac{C}{C_0} = 1 + \left(\frac{U_0}{U} - 1\right)\left(1 - \frac{1}{1 + (2 + 0.62\, U/U_0)\, Stk}\right), \tag{2}$$

with the Stokes number $Stk$ being defined by

$$Stk = \frac{\tau U_0}{D_s}, \tag{3}$$

and the relaxation time $\tau$ being

$$\tau = \frac{\rho_p\, d^2\, C_C}{18\eta}. \tag{4}$$

Here $\rho_p$ is the particle density (estimated for dust as 2.6 g cm$^{-3}$), $d$ is the aerosol diameter, $C_c$ is the Cunningham correction factor and $\eta$ is the viscosity of the air.

Figure S2 and S3 present $C/C_0$ as calculated by Eq. 2 as function of particle size for the mean, maximum and minimum $Q$ and $U_0$ occurring during flights of Cruiser (Fig. S2) and Skywalker (Fig. S3) at an altitude of 2000 m. The maximum error is negligible for particles below 1 µm, and grows with increasing particle size up to around ± 30% for particles of 10 µm in diameter for Cruiser and up to ± 60% for Skywalker.

Table S1: Variation of sample flow $Q$ and airspeed $U_0$ during the campaign.

|  | Skywalker | Cruiser |
|---|---|---|
| $Q_{mean}$ [lpm] | 4.91 | |
| $Q_{min}$ [lpm] | 4.37 | |
| $Q_{max}$ [lpm] | 5.56 | |
| $U_{0,mean}$ [m s$^{-1}$] | 17.4 | 28.1 |
| $U_{0,min}$ [m s$^{-1}$] | 14.3 | 23.4 |
| $U_{0,max}$ [m s$^{-1}$] | 23.5 | 33.3 |

[Figure]

Figure S2: Aerosol number concentration ratio due to anisokinetic sampling effects as a function of particle diameter for Cruiser.

[Figure]

Figure S3: Aerosol number concentration ratio due to anisokinetic sampling effects as a function of particle diameter for Skywalker.

*Pg. 7, ln. 8-9: I think a very brief discussion of the limitations and possible caveats of the FRIDGEs measurement principles is pertinent here. While these are listed in Schrod (2016), a brief summary here would also be useful, as these are of course also central to this work.*
We added a brief summary of the limitations of the measurement principle at the end of section 2.4.

Pg. 7, ln. 8-9 now read:

For a detailed description of the sampling procedure and FRIDGE's measurement principle as well as its limitations and possible caveats, see Schrod et al. (2016). These limitations include for example a) the possible loss of volatile aerosol constituents due to the analysis under medium vacuum, b) the possibility of a transient depletion of water vapor above the nucleating particles due to the uptake of water occurring at very high numbers of particles on the substrate, and c) technical restrictions regarding the method's time resolution. Although our measurements can cover the freezing induced by nuclei that are immersed in droplets after condensation (i.e. condensation freezing), they do not involve freezing of macroscopic droplets with immersed INPs.

*Pg.11, ln19: A brief mention as to what the physical meaning of the calibration factor, cf, is would be useful to the reader here.*
The calibration factor *cf* introduced by DeMott et al. (2015) has no deeper underlying physical meaning, but is just a fit parameter that refers to special instrumental conditions applied in the CFDC, i.e. $RH_w$ = 105%. In this manuscript, we use *cf* simply as a mathematical degree of freedom when fitting the observed measurements to the predicted INP concentrations.

We will add this paragraph to the manuscript.

Pg.11, ln.19 now reads:

The parameters α = 0, β = 1.25, γ = 0.46 and δ = −11.6 were empirically fitted by DeMott et al. (2015).  The calibration factor *cf* has no deeper underlying physical meaning. DeMott et al. (2015) state that the constants α, β, γ, and δ could have captured this coefficient, but they wanted to introduce it separately to account for instrument specific calibration of their CFDC. More precisely, they have found when measuring mineral dust aerosol in the AIDA cloud expansion chamber at $RH_{water}$ = 105 % a factor of 3 lower INP concentrations than the maximum concentration shortly before droplet breakthrough was observed in the CFDC (usually at $RH_{water}$ = 108 - 109 %). Therefore, they argue that a pre-factor of *cf* = 3 is needed to obtain the maximum immersion freezing INP concentration for mineral dust specific atmospheric data.
In this manuscript the calibration factor *cf* is handled completely independent of this definition. Instead, we use it simply as a mathematical degree of freedom when fitting the observed measurements to the predicted INP concentrations.

*Pg. 12, ln. 26-29: Can you show this good agreement by using a plot?*
See next answer.

*Section 3.2.2.: As you have ns values, it might be interesting to compare your results to the lab-based parameterisation for the ice nucleating activity of mineral dusts developed by Niemand et al. (2012) and maybe even to that for feldspar by Atkinson et al. (2013), taking into account that only a fraction of the dusts are likely to be feldspar.*

We will add a plot (Fig. S5) in the supplement that compares the measurements of this study to the data of Boose et al. (2016) and the parameterizations of Niemand et al. (2012) (hereafter: N12) and Atkinson et al. (2013) (hereafter: A13).

Our data agree well to the field data of Boose et al. (2016), but are lower than the lab data. The data agree acceptably with N12, where the highest measured $n_s$ is usually within about one order of magnitude of N12 or better. However, the K-Feldspar parameterization A13 does not match the observed slope. Especially for cold temperatures the data diverge from A13 by several orders of magnitude. As the referee pointed out, probably only a fraction of atmospheric dust particles is composed of this highly ice active material, so we believe this to be a difficult comparison to make. It is noteworthy that these two parameterizations are purely dependent on temperature, whereas we find our data to be dependent on $RH_{ice}$.

We will add a text passage to the manuscript.

Pg.12, ln.24 and following now read:

The values of AF and $n_s$ compare reasonably well with  measurements performed in  atmospheric environments influenced by mineral dust. E.g. Boose et al. (2016) found the active site density  from two month measurement data  at Izaña in Tenerife to range between $7\times10^7 - 3\times10^8$ m$^{-2}$ at T = −25 °C, $RH_{ice}$ = 130% (this study: $2\times10^7 - 7\times10^8$ m$^{-2}$ at T = −25 °C, $RH_{ice}$ = 129%, Fig. 5a in the supplement) and between $2\times10^8 - 2\times10^9$ m$^{-2}$ at T = −33 °C, $RH_{ice}$ = 135% (this study: $7\times10^7 - 2\times10^9$ m$^{-2}$ at T = −30 °C, $RH_{ice}$ = 135%, Fig. S5b in the supplement). Figure S5 also compares the observed active site densities with the laboratory based mineral dust immersion freezing parameterizations of Niemand et al. (2012) (hereafter: N12) and Atkinson et al. (2013) (hereafter: A13). Both parameterizations predict higher active site densities than were found in this study. The data agree acceptably to N12, with the highest measured $n_s$ being usually within the same order of magnitude as N12, or better. However, the K-Feldspar parameterization A13 does not match the observed slope. Especially for cold temperatures the data diverge from A13 by several orders of magnitude. As probably only a fraction of dust particles was composed of this highly ice active material, we did not expect a good agreement.

The following will be added to the supplement:

[Figure]

Figure S5: IN active site densities measured in the atmosphere and on mineral dust test aerosols. This study (boxes and diamonds); Boose et al. (2016) (circles); N12 (triangles and red line); A13 (hexagons and black line); dashed lines: water saturation at stated temperature.

*Pg.15, ln. 1: It should be noted here that Conen et al. examined the ice nucleating activities of dusts at much warmer temperatures than were probed here. That being said, a similar point is made by Tobo et al. (2014), which might be good to reference here.*
We added the temperature range in the sentence about Conen et al. (2011) and included a reference to Tobo et al. (2014).

Pg.15, ln. 1 now reads:

For example, Conen et al. (2011)  concluded from soil dust measurements in the range of −4 °C to −15 °C that the carbon content/biological residues within dust samples can define their ice nucleation properties. Similarly, Tobo et al. (2014) underlined the significance of organic matter in soil dusts as INPs in mixed-phase clouds at temperatures warmer than −36 °C.

References:

Atkinson, J. D., Murray, B. J., Woodhouse, M. T., Whale, T. F., Baustian, K. J., Carslaw, K. S., Dobbie, S., O'Sullivan, D. and Malkin, T. L.: The importance of feldspar for ice nucleation by mineral dust in mixed-phase clouds., Nature, 498(7454), 355–8, doi:10.1038/nature12278, 2013.

Belyaev, S.P. and Levin, L. M.: Techniques for collection of representative aerosol samples Journal of Aerosol Science, 5, doi: http://dx.doi.org/10.1016/0021-8502(74)90130-X, 1974.

Boose, Y., Sierau, B., arcía, M. I., Rodríguez, S., Alastuey, A., Linke, C., Schnaiter, M., Kupiszewski, P., Kanji, Z. A. and Lohmann, U.: Ice nucleating particles in the Saharan Air Layer, Atmospheric Chemistry and Physics, 16, 14, 9067–9087, doi:10.5194/acp-16-9067-2016, 2016.

Conen, F., Morris, C. E., Leifeld, J., Yakutin, M. V., and Alewell, C.: Biological residues define the ice nucleation properties of soil dust, Atmos. Chem. Phys., 11, 9643–9648, doi:10.5194/acp-11-9643-2011, 2011.

DeMott, P. J., Prenni, A. J., McMeeking, G. R., Sullivan, R. C., Petters, M. D., Tobo, Y., Niemand, M., Möhler, O., Snider, J. R., Wang, Z., and Kreidenweis, S. M.: Integrating laboratory and field data to quantify the immersion freezing ice nucleation activity of mineral dust particles, Atmos. Chem. Phys., 15, 393–409, doi:10.5194/acp-15-393-2015, 2015.

Hinds, W.C.: Aerosol Technology: Properties, Behaviors, and Measurement of Airborne Particles, Second Edition, John Wiley & Sons, Inc., New York, USA, 1999.

Niemand, M., Möhler, O., Vogel, B., Vogel, H., Hoose, C., Connolly, P., Klein, H., Bingemer, H., DeMott, P. and Skrotzki, J.: A particle-surface-area-based parameterization of immersion freezing on desert dust particles, J. Atmos. Sci., (2012), 2012.

Tobo, Y., DeMott, P. J., Hill, T. C. J., Prenni, A. J., Swoboda-Colberg, N. G., Franc, G. D. and Kreidenweis, S. M.: Organic matter matters for ice nuclei of agricultural soil origin, Atmos. Chem. Phys., 14(16), 8521–8531, doi:10.5194/acp-14-8521-2014, 2014.

Schrod, J., Danielczok, A.,Weber, D., Ebert, M., Thomson, E. S., and Bingemer, H. G.: Re-evaluating the Frankfurt isothermal static diffusion chamber for ice nucleation, Atmos. Meas. Tech., 9, 1313–1324, doi:10.5194/amt-9-1313-2016, 2016.

---

## Author Comment (AC2) · 28 Feb 2017

*This manuscript reports on a new technique, determining ice nucleating particle concentrations at various altitudes from drones. The technique clearly shows promise, and the measurements are needed in this area. I have a few questions/concerns, which the authors need to address prior to publication. Then, the manuscript will make a solid contribution to the field.*

***Major Comments:***

1. *The abstract ends with a bold and interesting conclusion that ground level INP measurements are of limited use in understanding INP aloft and their role in cloud formation. This may be true, but it is not substantiated by the data as presented. More could be done with the collected data set, as I describe below.*
   We think the data presented allow drawing a conclusion such as we have done. We will elaborate more on this matter as we go along answering the questions below.

2. *Pg. 5. The cruiser has a 2-stroke engine. 2 stroke engines are notoriously dirty running, producing significant pollutants, including NOX and likely particulates as well. There is a serious concern that the emissions from the engine will be active INP and thus will contaminate the INP sample and bias the reported concentrations. What has been done to check, correct and/or avoid this?*
   We shared the referee's concern. In order to identify any potential contamination, aerosol absorption was simultaneously monitored using a micro aethalometer (AethLabs, Model AE51) during each flight of Cruiser. The aethalomter's inlet was approximately 7 cm away from that of the INP sampler, ensuring that both instruments sampled the same air masses. Measurements from the aethalometer during each sampling are now provided in the supplement (Fig. S1). Although considerably noisy, these data do not show any significant large spikes that indicate particulate contamination.
   Furthermore the OPC record along the UAS flight track (e.g. Fig. 16) does not indicate any enhancement of particles in the downwind sectors of the spiraling path as compared to the upwind sectors (We now have added he 3000 m wind at 6 UTC from the DREAM model to Fig. 16). Likewise, this is true for the measurements of the aethalometer.

Additional evidence for the absence of severe contamination by engine exhaust comes from the analysis of individual particles on the Si substrates by electron microscopy/EDX. Only a very small fraction of 1-1.5% of the particles (in sample 25 of Fig. 18 / Tab. 3) was carbonaceous, which is consistent with the average absorption levels in the area (black carbon concentration about 0.5 µg m$^{-3}$). A similar result was found for a sample obtained with the battery-powered Skywalker x8 (sample 39 in Fig. 18 / Tab. 3).

Moreover, both UAS types were used alternatingly during the first half of the campaign and their INP concentrations show no significant differences (Fig. 10, Cruiser: diamonds, Skywalker: squares).

In summary, both methods employed did not identify any contamination suggesting that either the samples were free of exhaust particles or that the effect did not have an impact on the results presented in this work or the conclusions drawn from them.

Nevertheless, we will add a short remark about this important issue to the revised manuscript. The purple text above will be added to the supplement.

Pg. 5, ln. 29 and following now read:

The Cruiser (Fig. 6) is a fixed-wing, medium-size UAS (3.8m wingspan) with a two-stroke engine and a maximum take-off weight of 40 kg that can carry a payload of up to 10 kg for a maximum flight duration of 3 hours. Since this type of engine may produce a significant source of particle contamination, we thoroughly checked the data of an integrated aethalometer (AethLabs, Model AE51, Fig. S1 in supplement) as well as the data from electron microscopy (cf. section 3.4) for any indications of contamination, but did not find any evidence of contaminants in our samples. A small fraction of carbonaceous particles (<1.5%) was indeed identified in the samples. However, the same amount was also found in a sample acquired using the battery powered UAS, suggesting that their origin was not due to the engine's exhaust (Fig. 18 / Tab. 3).

[Figure]

Figure S1: Ambient aerosol absorption during INP sampling onboard the Cruiser. The raw output (black dots) and a rolling average (red line, based on a modification method of Hagler et al. (2011)) are shown. No indication of contamination by the two stroke engine's exhaust was identified.

3. *pg. 6 ln 27. Sampling times and therefore volumes vary by a factor of 3. Why wasn't the sample time kept uniform? What impact does this have on INP concentration results? This needs to be addressed.*

We adjusted the sampling times for each flight according to the dust forecast and current lidar images for each flight. When a heavy dust load was predicted, we scheduled a shorter sampling. The proper loading of a sampling substrate is crucial for the analysis in FRIDGE, with the targeted number of ice crystals on a substrate being between the lower limit of detection (defined by background noise) and an upper threshold (around 1000) above which crystals merge and are miscounted. One sample that was overladen needed to be discarded at certain measurement conditions in this work. We have no indication that the sample volume affected the measurement in the data presented here, since the activated fraction is uncorrelated to the sample volume. However, a volume effect can be found for laboratory samples heavily loaded with highly active aerosol.

4. *pg 10, ln 30- pg 11 ln 7 and Figures 11&12* (probably 10&12?): *The manuscript states the INP concentration is highly correlated to the concentration of large particle measured by the OPC and with the vertically integrated aerosol optical depth. Unfortunately, not all the data presented supports this conclusion. - This is the major issue with the manuscript.*

The stated high correlation refers to the data of Figs. 10 (and 12) that cover the entire length of the campaign. This statement is backed by the correlation coefficients of Tab. 2. We clarified now in the manuscript to what data we refer. Off course, not every single data point may be explained perfectly; however, this is very rarely the case in this particular field of ice nucleation. Rather than addressing every single case (which is unfortunately rather difficult as we will explain below), we chose a more general approach focusing on statistical averages and hoped to draw the attention of the interested audience to this new technique.

*In Figure 11, we see that INP correlations were essentially unchanged over the vertical altitudes sampled. (Particle concentrations are not included in the figure, but it is highly unlikely that they are equally invariable). When all the data is combined into one plot, interesting details are often lost, and I suspect that that is the problem here.*

We agree with the reviewer that interesting details may be lost upon averaging and pooling of data. However, our data coverage during individual flights is mostly too scarce for any interpretation (many flights collected only one sample). The only exception is April 9 (6 samples in total), which we present in the case study in chapter 3.3.

Although the statistical significance of the UAS-INP-profile might not be given, the median vertical profile shows the lowest INP concentration close to the surface with a gradual increase towards the top layers. More prominently, we found the ground INP concentration at the nearby CAO to be about one magnitude lower on average.

Unfortunately, we do not have a sufficient number of airborne aerosol observations to make a valuable addition to Fig. 11. However, the campaign-mean vertical profile of the volume depolarization ratio and DREAM dust mass concentration is now provided in Fig. S4. On average, a prominent vertical dust profile is visible. Yet, the single data points at the sampling time/altitude show differences of several magnitudes, which is a similar result to our INP measurements.

[Figure]

Figure S4: Left: Prominent vertical dust profile, when averaging over the whole campaign. Right: Large spread of single data points during INP sampling. Orange: DREAM dust mass concentration. Purple: Lidar volume depolarization ratio.

*Aren't there cases in which a dust layer aloft was sampled? Did INP concentrations increase with the dust layer? Yes or no? Perhaps dust layers were too thin to have an impact on the concentration of INP in any given sample? Or, did the drone miss the layers?*

We believe that we present sufficient proof in the manuscript that we were able to successfully sample from specific dust layers. We present evidence in our case study (section 3.3, Figs. 14−18 and Tab. 3) that the INP concentration increased in the dust layer. Here, we quote multiple times that INP concentrations were the highest of the campaign amidst this heavy dust layer (see pg.13, ln.4, pg.13, ln.33 and following, also visible in Figs. 9, 10, 12). We have now added the INP concentration measured from the ground station at Agia Marina to Fig. 17c, which will hopefully make this point more clear.

*Alternatively, what about contamination from the drone exhaust? Is that somehow causing falsely high values at lower altitudes (i.e. take-off and lands)?*

We refer to our answer of the comment 2.

*Rather than look at overall campaign correlations (i.e. Figure 12), it should be much more telling to look at cases. Likewise, correlations with particle counts are expected to vary vertically, especially when dust layers aloft are an order of magnitude higher than at ground level. The lidar is evidence that that as occurred in some cases here, but the connection to INP concentration is not demonstrated.*

We agree with the reviewer on the potential of case studies, with the caveat that sufficient data must be available. When this is given, like in section 3.3, we demonstrate the link between particle concentration and INP (in Fig. 17).

Regarding the particle concentration aloft we can say that during the major dust event particle number concentration measured in flight by OPC was about a factor of 3 higher than the maximum concentration at the surface a couple of hours later. Similarly, $PM_{10}$ at the surface was a factor of 4 lower than the DREAM dust mass prediction aloft.

The connection between the lidar measurements and INP (and aerosol) concentration is hinted at in Fig. 17c. Here the blue line shows the lidar-retrieved concentration of aerosol particles with d>0.5 µm (lower scale) as well as the INP concentration derived from it using the parameterization of Fig. 12d (upper scale). The black line shows the concentration of particles with d>0.5µm as measured by OPC onboard (lower scale) as well as the INP concentration derived from it (upper scale). The lidar-retrieval agrees with the OPC concentration (lower x-axis) as well as the INP measurements (upper x-axis).

Furthermore, we will demonstrate the connection between the lidar measurements and the INP concentration by calculating the correlation between the two. The correlation between the volume depolarization ratio and the INP concentration at T = -30 °C and $RH_{ice}$ = 135.4% is R = 0.74 (N = 46). We will add a sentence to the discussion of the results and include this finding in Tab. 2.

Pg.10, ln.29 – Pg.11, ln.2 now read:

The dominance of large scale dust advection can be seen from the correlation between the levels of INP aloft and at the ground (Tab. 2). The highest correlation is found between INP and the total particle number concentration with diameters larger than 0.5 µm ($n_{a>0.5}$) both measured on board the UAS (R = 0.97, n = 11). The volume depolarization ratio at the time and altitude of the INP sampling is also well correlated (R = 0.74, n = 46). The correlation between the individual local concentrations of INP sampled from UAS and of the aerosol mass concentration calculated for the same sampling path by the DREAM model is R = 0.69, n = 49. INP from the UAS are correlated to coarse mode PM (R = 0.59, n = 49) measured at CAO at ground level and to the vertically integrated AOT (R = 0.31, n = 49). Furthermore, the peaks of INP and the mineral dust parameters coincide (Fig. 10).

*In Figure 14, where a case study is presented, peaks in INP concentrations do not coincide with peaks in backscatter coefficient, especially the point at 10:00 UTC of 180 INP/std l. Any idea what happened at this point?*

We think that most (but not all) of the INP concentrations appear to be in reasonable agreement with the lidar backscatter. However, we agree with the reviewer that the discrepancy between sample #28 in Fig. 14 (180 INP/L) and the LIDAR backscatter measurements is disturbing. We have currently no satisfying explanation and can only speculate. We address this matter on pg. 13, ln. 14-21, to where we added a line to the manuscript.

Pg.13, ln.18 and following now read:

The depolarization signal (Fig. 14b), on the other hand, showed still a signal of medium to high strength for a broad range of altitudes up to 3km, suggesting that a considerable amount of mineral dust might have been collected. In fact, no depolarization ratio from any other day corresponding to the time/altitude of the samplings was found to be higher. Furthermore, we cannot rule out that the observed differences might have been caused by a heterogeneity in the dust spatial distribution between the two different operational sites.

*Backscattering coefficient can be complicated by many particle characteristics. – It would be nice to also include the OPC concentrations along with the INP concentrations.*
OPC concentrations are not included in Fig. 14, because they were only available for the flight that is marked with the red rectangle (samples 24 and 25). Due to technical difficulties no data were available for the sampling period of the second Cruiser flight, and the Skywalker was not equipped with an aerosol monitor. We point the referee to the Figs. 16 and 17c,d and section 3.3, where the OPC data of samples 24 and 25 are presented and discussed.

*Also, in 14B depolarization ratio and INP are not correlated in any way.*
We want to point out that the numbers appearing in Fig. 14b are not the INP concentration, but the sample identification numbers as it is indicated in the caption (if there was any confusion about this). As discussed above, we find overall a good correlation between the volume depolarization ratio and INP. Furthermore, we consider the agreement of depolarization ratio and INP for the case study in section 3.3 to be at least reasonable.

*In summary, the vertical profile of INP may be contaminated by the drone's engine exhaust? And INP concentrations may or may not be sensitive to cases of dust events. Given the wealth of data collected here including all the key elements to really look at dust, dust size, and INP at multiple altitudes, I urge the authors to consider additional cases are available which support their statement that INP is highly correlated to large particles. If the cases do not support that correlation, other interpretations of the data should be considered.*
We support the referee's vision of a more specific case-by-case study to gain a detailed understanding of the relationship between mineral dust and INP. Nevertheless, we hope that we now have eradicated the referee's concerns and that the referee now understands why a case-by-case approach was just not feasible in this study. We need to stress here that the data coverage in terms of time resolution is limited with the FRIDGE instrument, as compared to a CFDC. Often there were only two or less data points generated per day (with 6 samples being the most on the presented day of April 9). Of these few samples we have only one respective INP concentration value (each for multiple combinations of T and RH) that corresponds to a spatially and temporally integrated average of the sampling path.
We agree that there is room to improve for future campaigns with the combination of FRIDGE and UAS. Still, we believe that the data itself and the way it is presented are substantial enough to allow us to draw the conclusions we did.

5. *On figure 12, it is impressive that the predicted numbers are lower than measurements in 12a and 12b, but well correlated in 12c&d. It would be nice to expand the discussion of how these were parameterized differently, as it seems that this result is a key finding of the manuscript.*

We need to correct the referee here. As it can be seen in Fig. 12 and read on P.11 l.26-27 and P.12 l.1-3 the predicted concentrations for D10 and D15 were found to be higher than the measurements (about one magnitude for D15).

However, while reading the given lines again, we admit that the phrasing of the statement on P.11 l26-27 is in fact misleading. We apologize and rephrase this line.

P.11 l26-27 now reads:

While the slope of Fig. 12b is close to unity, the  measurements are  one order of magnitude lower than the estimate based on the parameterization .

In the paragraph following this line we list possible explanations for this offset.

References

Hagler, G. S. W., Yelverton, T. L. B., Vedantham, R., Hansen, A. D. A., and Turner, J. R.: Post-processing Method to Reduce Noise while Preserving High Time Resolution in Aethalometer Real-time Black Carbon Data, Aerosol and Air Quality Research, 11, 539–546, doi:10.4209/aaqr.2011.05.0055, 2011.